# Bacterial population dynamics during colonization of solid tumors

Serkan Sayin [1,4], Motasem ElGamel [2,4], Brittany Rosener [1], Michael Brehm [3], Andrew Mugler [2✉]
& Amir Mitchell [1✉]

## Abstract

**Bacterial colonization of tumors is widespread, yet the dynamics during colonization remain underexplored. Here we discover strong variability in the sizes of intratumor bacterial clones and use this variability to infer the mechanisms of colonization. We monitored bacterial population dynamics in murine tumors after introducing millions of genetically barcoded *Escherichia coli* cells. Results from intravenous injection revealed that roughly a hundred bacteria seeded a tumor and that colonizers underwent rapid, yet highly nonuniform growth. Within a day, bacteria reached a steady-state and then sustained load and clone diversity. Intratumor injections, circumventing colonization bottlenecks, revealed that the nonuniformity persists and that the sizes of bacterial progenies followed a scale-free distribution. Theory suggested that our observations are compatible with a growth model constrained by a local niche load, global resource competition, and noise. Our work provides the first dynamical model of tumor colonization and may allow distinguishing genuine tumor microbiomes from contamination.**

**Keywords** Tumor; Colonization; Microbiome; *E. coli*; Zipf's Law
**Subject Categories** Computational Biology; Microbiology, Virology & Host Pathogen Interaction

## Introduction

The human microbiota that reside within different organs, such as the skin, mouth and gut, are key players that can affect cancer progression and influence treatment success (Hanahan, 2022; Helmink et al, 2019; Zhao et al, 2023). Research over the past decade suggested that solid tumors can frequently harbor their own microbiome (Sepich-Poore et al, 2021; Ma et al, 2020; Riquelme et al, 2019). Recent surveys revealed that two-thirds of breast and pancreatic tumors harbor intratumor bacteria and that even tumors developing in sterile body sites, such as the brain and bones, frequently harbor a microbiome (Nejman et al, 2020). Bacterial

widespread presence in tumors is attributed to opportunistic colonization of the immunosuppressive microenvironment that exists after solid tumors have already formed (Sepich-Poore et al, 2021; Jiang et al, 2023). Notably, very recent publications argued against some studies in the field, such as those detecting bacterial DNA signature in patient blood samples (Gihawi et al, 2023; Poore et al, 2020, 2024; Sepich-Poore et al, 2024) and claims of fungal microbiome involvement in pancreatic cancer (Fletcher et al, 2023; Aykut et al, 2019). Interest in intratumor bacteria in recent years has also surged, with multiple efforts to intentionally administer to patients genetically engineered bacteria for the detection and treatment of solid tumors (Gurbatri et al, 2022; Lu and Tong, 2024).

While the significance of the tumor microbiome has promoted research in this field, it has left fundamental questions on the biological processes underlying this phenomenon underexplored. A key gap in knowledge concerns the process of tumor colonization. Studies of human patients typically aim to uncover evidence of bacterial presence in cancer tumors by inspecting tumors and healthy tissues that were removed during surgery. Common methodologies are quantitative PCR for the detection of microbial DNA or microscopy imaging of tumor sections with stains or antibodies that are unique for microbial components, such as the bacterial lipopolysaccharides (e.g., Senthakumaran et al, 2024; Geller et al, 2017). While these end-point measurements can provide extreme sensitivity for detecting microbial presence, they provide little information about the temporal dynamics of colonization.

The suggested models on tumor colonization rely on the observed overlap between species composition in specific "natural" microbiome sites and the species identified in tumors in specific body sites (Bullman et al, 2017; Boesch et al, 2022) and suggest translocation events of bacteria between organs (Schorr et al, 2023) or into the blood stream (Cummins and Tangney, 2013). Yet little has been empirically determined in human patients beyond this association. It therefore remains unclear how bacteria initially colonize the tumor niche and how a multi-species microbiome emerges (Goubet, 2023). It remains unknown if multi-species colonization arises through simultaneous seeding by multiple species or whether colonization follows a sequential seeding process reminiscent of gut colonization during development. Lastly, little is known about the population turnover of intratumor bacteria, including the growth and death rate of colonizing bacteria and whether population bottlenecks exist in different stages of colonization.

[1]Department of Systems Biology, University of Massachusetts Chan Medical School, Worcester, MA 01655, USA. [2]Department of Physics and Astronomy, University of Pittsburgh, Pittsburgh, PA 15260, USA. [3]Program in Molecular Medicine, University of Massachusetts Chan Medical School, Worcester, MA 01655, USA. [4]These authors contributed equally: Serkan Sayin, Motasem ElGamel. ✉E-mail: andrew.mugler@pitt.edu; amir.mitchell@umassmed.edu

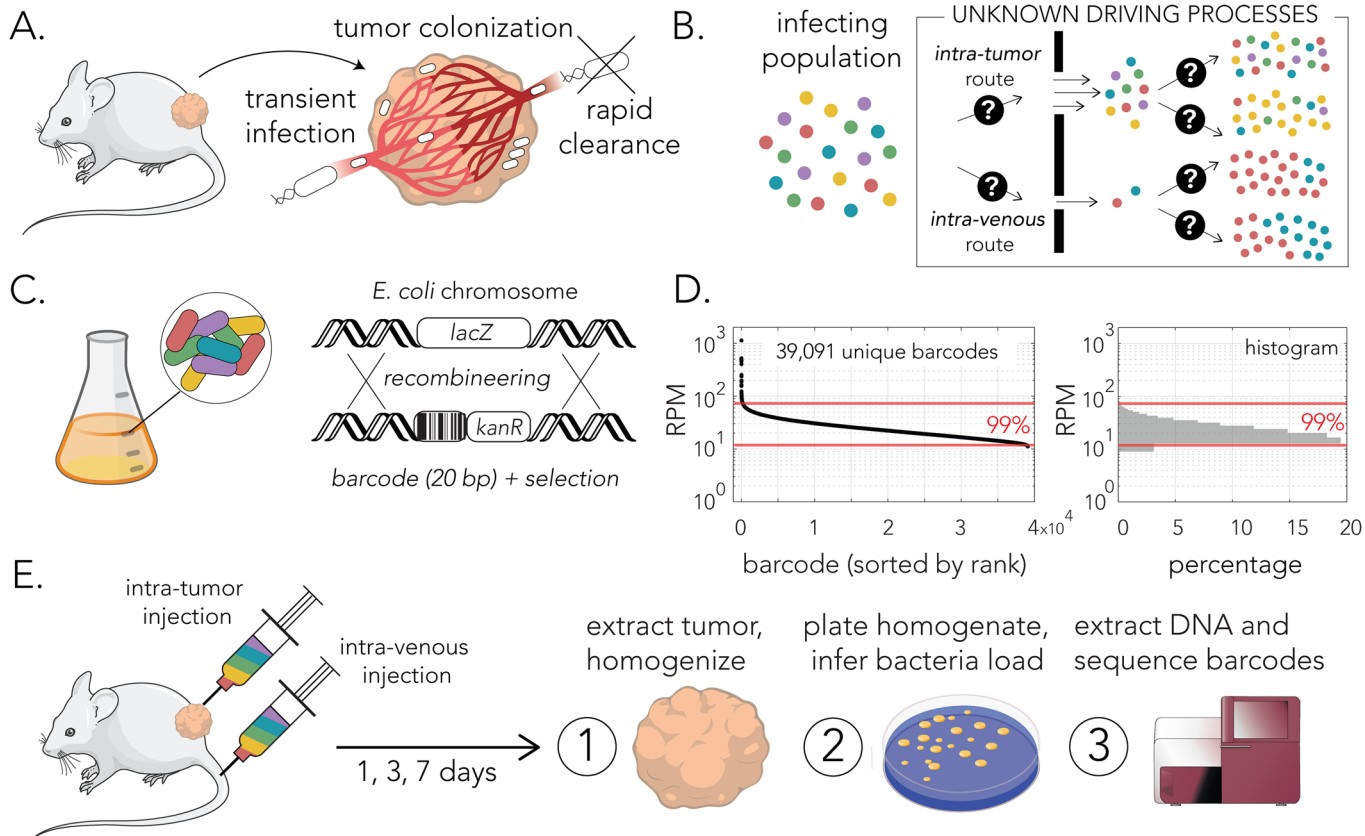

**Figure 1. Monitoring bacterial population dynamics in a syngeneic mouse tumor model.**

(A) Colonization of subcutaneous murine tumors in immunocompetent mice is achieved by transient systemic infection. Bacteria reaching the tumor niche can proliferate and persist due to limited access of the immune system. (B) Alternative modes of bacterial infection and growth in the tumor niche can be evident by clone diversity in resected tumors. (C) The design of the barcoded *E. coli* clone collection. A kanamycin resistance cassette with an upstream barcode region was integrated into the *lacZ* gene locus. (D) Distribution and histogram of barcode frequencies used for mouse experiments. Red lines indicate the central 99% of the data points. (E) Outline of experimental approach. The clone collection was introduced by intravenous or intratumor injection (each mouse had two subcutaneous tumors). Mice were euthanized, and tumors were resected after 1, 3 or 7 days after infection. Tumor homogenates were used to calculate bacterial load and to extract DNA for deep sequencing of the barcode region. Data information: Panel (D) Biological replicates $n = 1$, Technical replicates $n = 4$, data from a single experiment passing the threshold and filtering criteria explained in Methods. Source data are available online for this figure.

Here we use an established mouse model of intratumor bacteria to explore the bacterial population dynamics in controlled experiments (Fig. 1A). We introduced genetically barcoded *E. coli* cells by intravenous or intratumor injections and monitored how the bacterial load and clone diversity changed over time (Fig. 1B). Our experiments revealed that most clones die out before the first day, indicating strong colonization bottlenecks. Yet, the surviving clones coexist persistently for the remainder of the experiment. In the intratumor case, we discovered a robust statistical pattern in clone abundances, known as Zipf's law: barcode abundance sharply diminishes inversely of its rank, regardless of the experiment or collection day. To understand the meaning of this pattern, we followed the approach of Luria and Delbrück, who famously used mathematical modeling to show that abundance statistics, rather than average dynamics, reveal the evolutionary mechanism of acquired bacterial resistance to viral infection (Luria and Delbrück, 1943). Our model draws from the classic frameworks of theoretical ecology that have been applied to microbial systems (Berg et al, 2022; Bunin, 2017; MacArthur, 1970; Chesson, 1990), and adapts them to the specifics of the tumor environment.

Taken together, our controlled experiments and theory provide the first dynamical model of tumor colonization and sustained diversity. Our results suggest that the mechanisms that give rise to the statistics of Zipf's law—local growth inhibition and strong growth noise—are the same mechanisms that allow many different strains of colonizing bacteria to coexist in a steady state. Our results shed light on the origin of tumor-microbiome diversity and suggest that statistics may allow distinguishing genuine tumor microbiomes from contamination.

## Results

### Monitoring population dynamics with an *E. coli* barcoded strain collection

We prepared a library of barcode-tagged clones using the *E. coli* Nissle 1917 strain (EcN). This commensal isolate, currently prescribed as a probiotic, is widely used for studying intratumor

bacteria in mouse models (Geller et al, 2017; Yu et al, 2019; Leventhal et al, 2020; Gentschev et al, 2022; Gurbatri et al, 2024; Wu et al, 2023). We prepared the strain collection by PCR amplifying a kanamycin resistance cassette with a forward primer that included an upstream region of twenty random nucleotides and integrating the amplicon into the *lacZ* gene locus (Fig. 1C). We pooled together roughly 50,000 individual colonies and grew them overnight before characterizing the barcode diversity. We extracted DNA from the culture and sequenced the barcode region in four technical replicates. We identified and enumerated the individual barcodes with the bartender software package (Zhao et al, 2017) and found 39,091 unique barcodes that met our conservative thresholds (Methods). Figure 1D shows the frequency of barcodes in our library ordered by their rank and normalized to a million reads (RPM). We observed that barcodes followed a narrow frequency distribution with 99% of barcodes represented within a six-fold frequency range. This clone collection was used for the mouse experiments in our work (Fig. 1E).

Even in a constant environment, clone proportions within the barcoded collection are expected to gradually change over time, despite all clones being equally fit, due to natural variability in growth and death rates in a finite population of cells. To evaluate the magnitude of this expected variability, which we term intrinsic noise, we inoculated the barcoded strain library into four different media types and allowed it to grow until reaching late-log phase (Fig. EV1A). Figure EV1B shows a comparison between the barcode frequencies before and after the experiment. Importantly, while we observed the expected variation in clone frequency after this simple growth experiment, we also noted that the overall distribution of clone frequencies was not altered by growth rate, as demonstrated by the histograms in Fig. EV1B. Mathematical analysis reveals that a Monod model (described later) with intrinsic noise captures behavior in all four growth conditions and can therefore be used as a null model for barcode variation in our barcoded strain collection.

## Narrow colonization bottleneck and rapid bacterial growth after intravenous injection

We studied bacterial tumor colonization in a syngeneic murine model previously used by others to study intratumor bacteria (Geller et al, 2017; Yu et al, 2019; Leventhal et al, 2020). Figure 2A outlines our experimental approach. First, tumors were seeded by subcutaneous injection of a mouse CT-26 cancer cell line into the left and right flanks of each immunocompetent mouse (BALB/c). Tumors were allowed to form over ten days before systemic bacterial administration. To allow controlled tumor colonization, the barcoded clone library was introduced by tail vein injection, and the mice were continuously monitored throughout the remainder of the experiment. Groups of four to five mice were euthanized after 1, 3, or 7 days post bacterial injection. Tumors were then removed, weighed, and homogenized. Lastly, the homogenized tissue was used to determine the bacterial load, by plating on selective agar plates, and to determine clone diversity by extracting DNA and deep sequencing of the barcode region.

A summary of the bacterial load and clone diversity is shown in Fig. 2B. Overall, we observed that rapid changes in bacterial load and clonal frequency took place within a day of injection and seemed to stabilize for the remainder of the experiment. We

detected roughly only a hundred unique barcodes in each of the tumors resected throughout the experiment (average barcode number was 52, 132, and 84, on days 1, 3, and 7, respectively). When comparing the clone identity in the same-mouse tumor pairs, we observed that the number of overlapping barcodes was higher than that expected by random in 8 out of 13 mice (Fisher's exact test with a Bonferroni correction, Dataset EV1). This overlap likely arises from the relatively rare migration of bacteria between tumors previously reported in this mouse model (Harimoto et al, 2022). However, overlapping barcode frequencies in all mice but one were not correlated across tumor pairs (Spearman rank correlation with a Bonferroni correction, Dataset EV1). We also did not observe significant correlation in barcode frequencies between mice or on the same flank side, except for two comparisons out of 1540 comparisons (Dataset EV4). Thus, tumors seem to have operated as separate and independent niches. We observed that the average bacteria load per tumor reached $\sim 5 \times 10^8$ cells within a day of injection and further increased to $\sim 4.7 \times 10^9$ cells by the third day, and then slightly decreased to $2.6 \times 10^9$ cells. Taken together, our data suggested that a very narrow bottleneck limits bacterial seeding of the tumor niche, yet once seeding occurs, bacteria rapidly grow and almost saturate the tumor niche within a day.

We tested if colonization statistics were correlated in the same-mouse tumor pairs across twelve mice from the entire experiment. Figure 2C shows the relationship between the number of detected barcodes. A Spearman rank correlation test suggested this correlation is not statistically significant ($p$ val = 0.22). Figure 2D shows the relationship between the number of detected barcodes and the tumor carrying capacity (bacteria per gram of tumor), which were not found to be correlated (Spearman rank correlation test, $p$ val = 0.90). Assuming that each detected barcode represents a successful colonization event, we can infer that the tumor bacterial load does not depend on the initial number of colonizers. Lastly, Fig. 2E shows the relationship between the carrying capacity across the same-mouse tumor pairs, which were observed to be correlated (Spearman rank correlation test, $p$ val = 5.38e-04). Taken together, these statistics suggest the following: bacteria colonized each of the tumors independently, and the number of bacteria seeding the tumor, assumed to equal to the number of detected barcodes, is independent of the bacterial load observed after colonization. However, the significant correlation in tumor bacterial carrying capacity suggested this characteristic is likely determined at the host level.

Finally, we evaluated the uniformity of bacterial load within each tumor by inspecting the size of each successful colonizing clone. Figure 2F shows results from a representative single mouse (euthanized a single day after bacteria injection). As the figure shows, we observed that clone sizes were highly nonuniform, with 54% of all bacteria in the right tumor originating from a single clone. The second most common clone contributed 14% of the bacterial load. Similar trends were observed across all tumors isolated after intravenous injections (Dataset EV2, Figure EV2A). We did not find any significant correlation between the barcode frequency in the inoculum and its frequency in the tumor (Fig. EV2B). We used Shannon entropy to quantify clone diversity across all days and compare it to the entropy of the injected clone library. Figure 2G shows the entropy calculated across all 27 tumors collected after intravenous injections. As the figure shows, all the

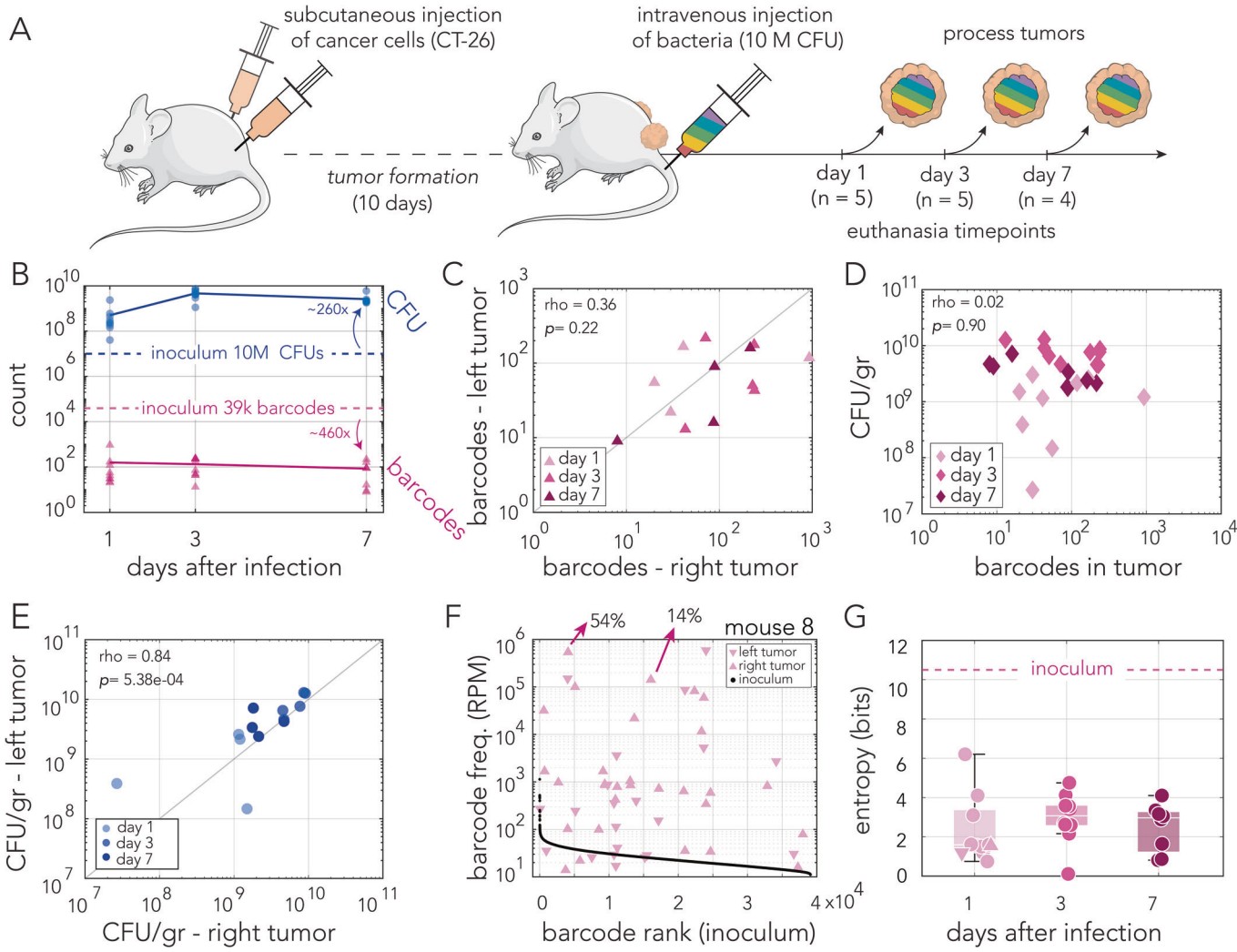

**Figure 2. Tumor colonization after intravenous injection.**

(A) Outline of experimental approach. Subcutaneous tumors were formed on the right and left flanks. After tumor formation, mice were infected intravenously by injection of ten million CFUs. Groups of four to five mice were euthanized on days 1, 3, and 7 post infection. Resected tumors were processed to determine bacterial load and clone diversity. (B) Bacterial load and number of detected barcodes in the tumors are shown in blue and pink color across three terminal timepoints, respectively. The number of clones in the inoculum are shown for comparison (dashed lines). Lines mark the average. (C) Scatter plot showing the number of barcodes detected in tumor pairs resected from individual mice. Spearman's rank correlation test, $p = 0.22$. One mouse was excluded since tumors did not develop on both flanks. (D) Scatter plot showing the number of detected unique barcodes from the tumors and the carrying capacity (CFU/gr tumor). Spearman's rank correlation test, $p = 0.90$. (E) Scatter plot showing the carrying capacity (CFU/gr tumor) in tumor pairs resected from individual mice. Spearman's Rank Correlation test, $p = 5.38e-04$. (F) Frequency of detected unique barcodes from both tumors of a representative mouse euthanized at day 1. Barcode frequency distribution of the inoculum is plotted for comparison (black dots). The two highest frequency barcodes found in the right tumor are shown with arrows to highlight that the top barcodes make up the majority of the bacterial population. (G) Box-whisker plot of Shannon entropy index of each tumor based on the number of unique barcodes detected. Data points from individual tumors are shown as circles (triangles match tumors from panel F). The Shannon entropy Index of the inoculum is shown for comparison (dashed line). Box: lower and upper quartiles, white line: median, whiskers: the most extreme data points excluding outliers. One-way ANOVA test shows that there are no significant differences across days ($F = 0.42$, $p = 0.6597$). Data information: Data shown from a single experiment. Number of total mice $n = 14$, Panels (B, D, E, G). Panel (C), $n = 13$. Panel (F), $n = 1$. Technical replicates $n = 1–2$ for Panel (B), $n = 2$ for Panels (C, D–G). Source data are available online for this figure.

tumors were characterized by a considerably lower entropy relative to the injected library. This lower entropy reflects both the overall low number of barcodes as well as their nonuniform clone sizes. After the initial drop, entropy remained stable across all days (one-way ANOVA, $p$ val = 0.66).

In summary, measurements of bacterial load and clone diversity in tumors colonized after intravenous injection of bacteria revealed that roughly only a hundred individual bacteria successfully

colonized the tumor niche, yet they rapidly grew to almost reach load saturation within a single day. The relative size of each detected clone revealed that colonizing bacteria experience varying degrees of colonization success, with leading clones taking up over half of the total bacterial load. If we assume that the leading clone originated from a single cell, we can roughly estimate the bacterial growth rate within the tumor niche. If the most successful clone reaches a population size of $2.5 \times 10^8$ within a day of injection, it

divides roughly 28 times during that period (corresponding to a generation time of ~50 min).

## Highly uneven clone abundance matches Zipf's law after intratumor injection

We studied bacterial clone dynamics in a model that allows significantly wider tumor colonization bottlenecks by modifying the route of inoculation. In this model, we inoculated the mice with an identical number of bacteria from the barcoded collection by direct injection into the tumors (Fig. 3A). A summary of the bacterial load and clone diversity is shown in Fig. 3B. We detected thousands of unique barcodes in each of the tumors resected throughout the experiment (the average number of detected barcodes were 7238, 3460, and 3566, on days 1, 3, and 7, respectively). We did not detect a significant correlation in the frequency of overlapping barcodes identified in the same-mouse tumor pairs in any mouse (Spearman rank correlation with a Bonferroni correction, Dataset EV1). Thus, similarly to intravenous injection, tumor pairs in the intratumor injection model seem to have operated as separate and independent niches. Calculation of bacterial load showed a trend similar to that observed after intravenous injection. We observed that the average bacteria load per tumor reached ~$1.4 \times 10^9$ cells within a day of injection and further increased to $4.7 \times 10^9$ cells by the third day and then slightly decreased to $1.9 \times 10^9$ cells. A clear difference seen in the two inoculation models was in the number of detected barcodes. After intratumor injection, we detected almost two orders of magnitude more barcodes than we detected after intravenous injection.

We tested if the number of detected barcodes were correlated in the same-mouse tumor pairs across 15 mice from the entire experiment. Figure 3C shows the relationship between the number of detected barcodes. A Spearman rank correlation test suggested this correlation is not statistically significant ($p$ val = 0.7). Assuming that each detected barcode represents a successful colonization event, we can infer that the tumor bacterial load does not depend on the initial number of colonizers. Figure 3D shows the relationship between the number of detected barcodes and the tumor carrying capacity (bacteria per gram of tumor), which were not found to be correlated (Spearman rank correlation test, $p$ val = 0.54). Taken together, these statistics suggest that the identity of clones that successfully colonized the tumors is independent across the same-mouse tumor pairs and that the number of successful colonizing clones is independent of the total bacterial load. These trends are similar to those observed in the intravenous injection model.

Finally, we evaluated the uniformity of bacterial load within each tumor by inspecting the size of each successful colonizing clone. Figure 3E shows results from a representative single mouse (euthanized after a single day). As the figure shows, we observed that clone sizes were highly nonuniform, with 6.3% of all bacteria in the left tumor originating from a single clone. The second most common clone contributed 2.2% of the bacterial load. Similar trends were observed across all tumors isolated after intratumor injections (Dataset EV2; Fig. EV3A). We did not find any significant correlation between the barcode frequency in the inoculum and its frequency the tumor (Fig. EV3B). Calculation of Shannon entropy to quantify clone diversity across all days showed considerable entropy decrease relative to the injected clone library (Fig. 3F). However, entropy remained

stable across days (one-way ANOVA, $p$ val-0.36). To better evaluate the variability in clone representation, we decided to inspect the relative proportion of successfully colonizing clones in each tumor. Figure 3G shows the proportion of each clone within each tumor relative to its rank, presented on a logarithmic scale. Surprisingly, this analysis uncovered a linear relationship, with a slope of -1, between the logarithm of the clone rank and the logarithm of its proportion across the entire rank range. To further confirm this relationship, we fitted the rank ($r$) vs clone proportion ($f$) data to a nonlinear function of the form $f = \beta r^{-\alpha}$ for all experiments. We found that the best fit value of the exponent $\alpha$ across experiments has a median of 0.954 and a standard deviation of 0.558. Moreover, this quantitative relationship held across all tumors and all days of the experiment. This relationship revealed that clone sizes followed a characteristic power-law distribution termed Zipf's law (Newman, 2005).

In summary, measurement of tumors colonized by direct intratumor injections revealed shared trends across all days and mice: overall, a high number of unique barcodes, attributed to a considerably wider colonization bottleneck, and highly nonuniform representation across colonizing clones. The lack of uniformity allows ruling out the bacterial injection route as the cause for high variability in clone proportions, e.g., due to variable time of arrival to the niche. This observation suggests growth dynamics within the tumor likely underlie the high variability in clone proportions. Lastly, an inspection of the distribution in clone proportion uncovered that they follow a specific power-law distribution matching Zipf's law. This surprising trend was evident across tumor samples collected throughout the experiment.

## Whole-genome sequencing of dominant clones rejects clone adaptation as a driving force

Tumor takeover by tumor-specific clones evident by typically one or two dominant barcodes can be attributed to different processes, including selective advantage of some bacteria in the tumor niche due to spontaneously arising mutations. To test if dominant clones are associated with potentially advantageous mutations, we isolated individual clones directly from ten resected tumors on agar selective plates (Methods). We focused on tumors colonized after intravenous injections since the bacterial population expanded in them more than in intratumor injections. We sequenced the genomes of the ten isolated clones. Sequencing showed they all matched dominant barcodes, determined by independent amplicon sequencing, in their respective tumors. Eight of them represented the most frequent barcode, and two represented highly frequent, but not the most frequent, barcode. Seven of the clones were present in the 39,091 qualified barcode list and three were from barcodes detectable in our inoculum but with initial frequencies below our conservative threshold for qualified barcodes (Methods). Next, we compared the genomes of dominant isolates to each other to identify clone-specific mutations using the breseq tool (Deatherage and Barrick 2014). We did not identify any genetic differences among these ten clones (Dataset EV3) besides their unique individual barcodes. Given the *E. coli* spontaneous mutation rate, 0.001 genome/generation (Lee et al, 2012), and an expansion of a single cell to a population of $2 \times 10^9$ cells (above the maximal number of cells we observed in a tumor), there is 0.97 probability that a clone will be without any mutations. The lack of detectable mutations in dominant tumor clones rules out the

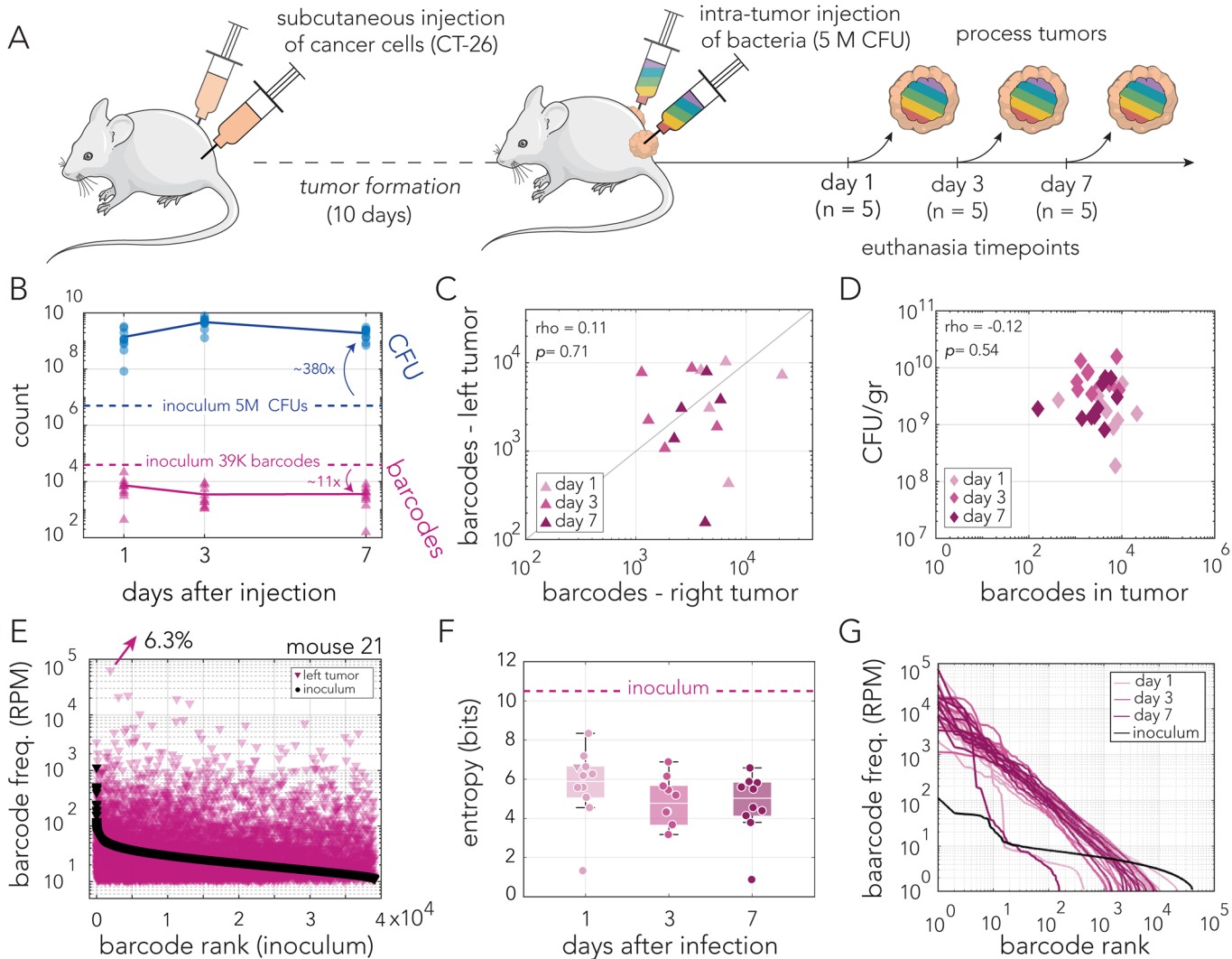

**Figure 3. Tumor colonization after intratumor injection.**

(A) Outline of experimental approach. Subcutaneous tumors were formed on the right and left flanks. After tumor formation, mice were infected by intratumor injection of five million CFUs per tumor. Groups of five mice were euthanized on days 1, 3, and 7 post infection. Resected tumors were processed to determine bacterial load and clone diversity. (B) Bacterial load and number of detected barcodes are shown in blue and pink color, respectively. The number of clones in the inoculum are shown for comparison (dashed lines). Lines mark the average. (C) Scatter plot showing the number of barcodes detected in tumor pairs resected from individual mice. Spearman's Rank Correlation test, $p = 0.70$. (D) Scatter plot showing the number of detected unique barcodes from the tumors and the carrying capacity (CFU/gr tumor). Spearman's rank correlation test, $p = 0.54$. (E) Frequency of detected unique barcodes from a single tumor of a representative mouse euthanized at day 1. A total of 10,258 unique barcodes were detected in this tumor. Barcode frequency distribution of the inoculum is plotted for comparison (black dots). The highest frequency barcode is shown with the arrow. (F) The Shannon entropy index of each tumor is based on the number of unique barcodes detected. Data points from individual tumors are shown as circles (the triangle matches the tumor from panel (E). Shannon entropy Index of the inoculum is shown for comparison (dashed line). Box: lower and upper quartiles, white line: median, whiskers: the most extreme data points excluding outliers. One-way ANOVA test shows that there are no significant differences across days ($F = 1.07$, $p = 0.3569$). (G) Ranked frequency distribution of the detected unique barcodes from each tumor in the experiment (shades of pink for days 1, 3, and 7) and the inoculum. Barcodes detected in each tumor or inoculum were ranked individually in descending order. Data information: Data shown from a single experiment. Number of total mice = 15 for panels (B–D, F, G). For panel (E) $n = 1$. Technical replicates $n = 1$–2 for Panel (B), $n = 2$ for Panels (C–G), $n = 4$ for inoculum at Panels (E, G). Source data are available online for this figure.

---

possibility that spontaneous adaptive mutations are a common mechanism underlying tumor takeover.

## Theoretical framework for tumor colonization

To quantitatively understand the growth dynamics and the emergence of Zipf's law, we developed a theory of bacterial colonization and growth in the tumor. We started with the simplest model that accounts for nutrient-limited growth—the Monod model—along with environmental noise. Then, motivated by the experimental observations, we added local growth limitation and a colonization bottleneck. We observed that these latter features were necessary to quantitatively match the experimental observations, including Zipf's law.

## Stochastic consumer-resource model of tumor colonization is insufficient

We began with the simplest consumer-resource model (Berg et al, 2022; MacArthur, 1970; Chesson, 1990) that encompasses competition for nutrients and environmental stochasticity:

$$\frac{dN_i}{dt} = \frac{gs}{s+k} N_i + \sigma \eta_i N_i \qquad (1)$$

$$\frac{ds}{dt} = -\gamma \sum_{i=1}^{B} \frac{gs}{s+k} N_i + D(s_0 - s) \qquad (2)$$

In Equations (Eq.) (1) and (2), $N_i$ represents the abundance of barcode $i$, and $s$ is the concentration of a shared resource within the tumor environment. For simplicity, we assumed that growth is limited by a single resource that is uniformly distributed within the tumor. The growth rate of each clone in Eq. (1) depends on $s$ via a Monod function (Monod, 1949), where $g$ is the maximum growth rate, and $k$ is the half-saturation constant. In turn, growth depletes the resource in Eq. (2), where $\gamma$ is the inverse yield coefficient, and $B$ is the number of surviving barcodes. We model nutrient consumption using a Monod function, as it was developed specifically for microbial populations, and we will show that it can be easily extended to account for local growth limitation (see next section). The resource is replenished (e.g., via passive diffusion into the tumor environment) at a rate $D$ from the rest of the surrounding healthy tissue with a background concentration $s_0$. Since all barcodes are genetically identical, we assumed that they share the same resource preferences and have the same fitness. Thus, model parameters are the same for all barcodes. Nevertheless, the individual growth rates of each barcode differ because of the noise term, described next.

The environmental stochasticity is modeled in Eq. (1) as multiplicative noise with strength $\sigma$, where $\eta_i$ is a barcode-specific white Gaussian noise term with zero mean and unit standard deviation. The noise term signifies environmental effects that are not explicitly modeled, such as fluctuations in access to resources, effects from the mouse's immune system, and other biotic and abiotic factors. Such multiplicative noise is typical in microbial ecology models (Grilli, 2020), as opposed to demographic noise, which has a different functional form and is due to the intrinsic dynamics of the system (Gillespie, 2000), or additive noise, which accounts for fluctuations in the migration rates of species (Descheemaeker and Buyl, 2020).

Numerical simulation of Eqs. (1) and (2) is shown in Fig. 4A, as a frequency-vs-rank plot (like Fig. 3G) at various times. Our simulations showed that the statistics do not converge to Zipf's law, and most barcodes eventually become extinct, i.e., the maximum rank continues to decrease over time. The reason for these dynamics is that several barcodes, selected at random due to the noise, come to dominate the system, leaving an insufficient amount of resources for the rest. We conclude that the model in Eqs. (1) and (2) cannot capture the experimental phenomena.

## Incorporating local growth limitation improves model fit with experiments

Preventing fast-growing clones from dominating the system generally requires a form of growth limitation that is dependent on the clone size $N_i$. The simplest form of growth limitation is to modify the Monod function to make the half-maximal constant proportional to the clone abundance ($k$ becomes $kN_i$):

$$\frac{dN_i}{dt} = \frac{gs}{s+kN_i} N_i + \sigma \eta_i N_i \qquad (3)$$

$$\frac{ds}{dt} = -\gamma \sum_{i=1}^{B} \frac{gs}{s+kN_i} N_i + D(s_0 - s) \qquad (4)$$

In fact, this modification is widely used and was first deduced from empirical observations of bacterial growth shortly after Monod's work, by Contois (Contois, 1959). Conceptually, this modification means that a larger population will grow at a smaller rate, e.g., due to crowding or any other effect that limit cells' access to nutrients within a larger population. Within our model, this modification assumes that a clone's growth is limited by its own abundance and not that of other clones. This assumption only makes sense if each clone exists in its own niche, e.g., if clones are spatially segregated. We rule out the possibility of global growth limitation (i.e., including $k\Sigma_i N_i$ in the denominator) since this is qualitatively equivalent to the Monod model and leads to population extinction. Nonetheless, barcodes still compete for shared resources at different rates depending on their size. We give experimental evidence for the spatial segregation assumption in the next section when we model the colonization bottleneck.

Numerical simulation of Eqs. (3) and (4) is shown in Fig. 4B (purple; see Methods for theoretical analysis, blue). We see that this model exhibits Zipf's law, consistent with the experimental data (Fig. 3G). This result demonstrates that the small modification to the Monod function in Eqs. (3) and (4) has significant implications for population stability and extinction prevention in the presence of stochasticity, which we expand upon further in the Discussion. Further incorporating the colonization bottleneck into the model (see next section) does not change the presence of Zipf's law, even as it decreases the number of surviving barcodes $B$ (Fig. 4C).

Next, we asked whether the model can capture the experimentally observed dynamics of the total CFU count and the number of surviving barcodes (Figs. 2B and 3B). Figure 4D,E shows the dynamics of our model for example parameter values. We see that, in general, the total CFU count increases and then saturates while the number of surviving barcodes decreases and then saturates, consistent with the experiments. These dynamics are tuned by specific model parameters: the final total CFU count is primarily determined by the ratio $s_0/\gamma$ (Fig. 4D), the probability that cells go extinct (as well as the amount of fluctuations) is primarily determined by the noise strength $\sigma$ (Fig. 4E), and the final number of surviving barcodes is primarily determined by the bottleneck size (Fig. 4F). These dependencies, discussed in more detail in the Methods, have intuitive explanations and allow us later to calibrate our model to the experimental data.

## Colonization bottleneck agrees with experimental colonization patterns

We modeled the bottleneck as a survival probability, $q$, for individual cells regardless of which barcode they belong to. Upon injection, each cell either establishes in the tumor and remains viable (survives) with probability $q$, or it does not (dies) with

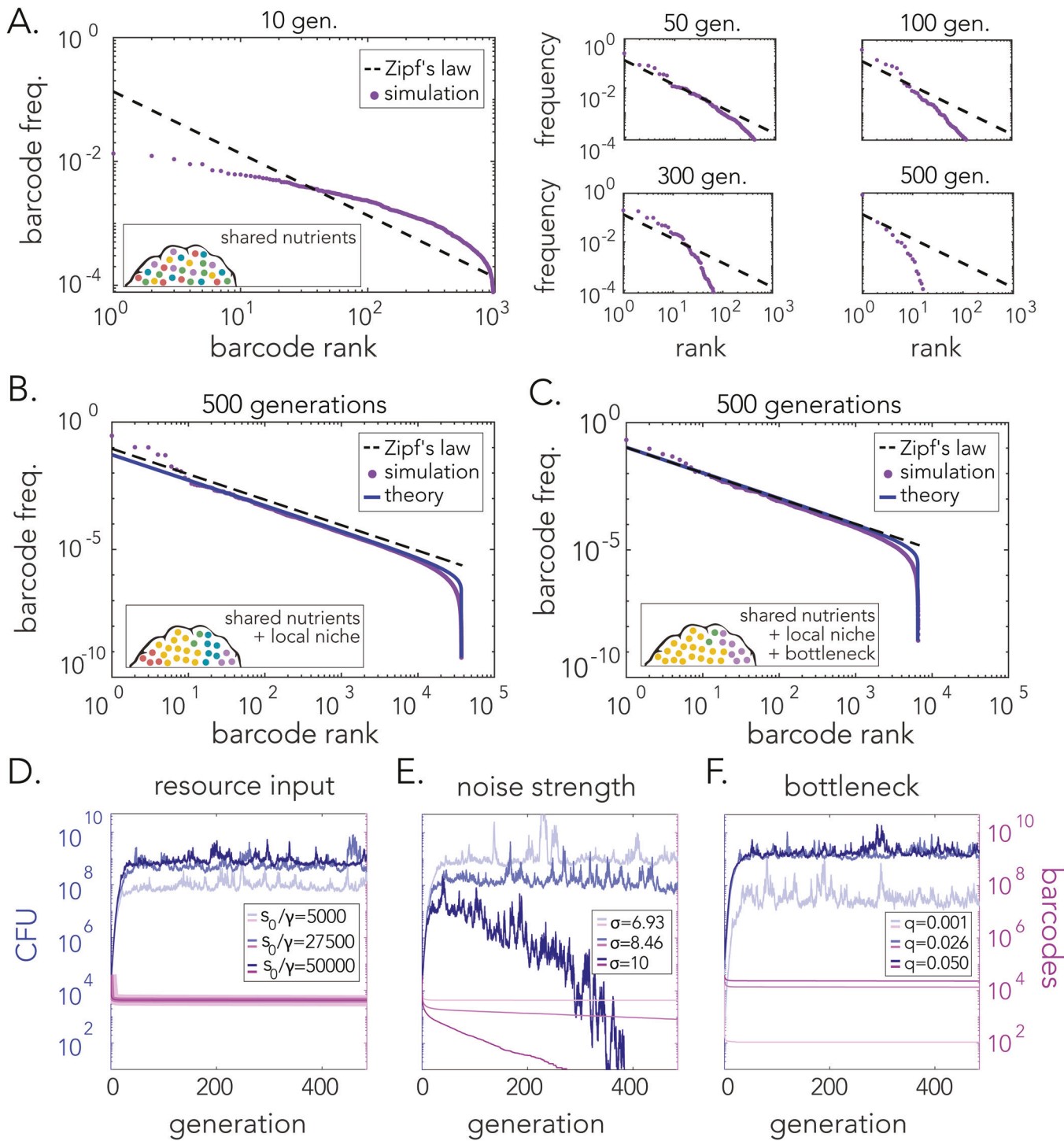

**Figure 4.** **(A)** Simulations of the stochastic consumer-resource model [Eqs. (1) and (2)] across different generations demonstrate deviation of the statistics from Zipf's law and the eventual extinction of the population after many generations. **(B)** Simulation of the model after incorporating local growth limitation [Eqs. (3) and (4)] demonstrate that the statistics satisfy Zipf's law even after 500 generations. **(C)** Same simulation as in B after the inclusion of an initial bottleneck leading to the extinction of some barcodes. **(D–F)** Illustrations of the effects different model parameters have on the dynamics of the total population CFU and the number of surviving barcodes. The ratio of introduced resources to the inverse yield coefficient, $s_0/\gamma$, affects the final CFU without affecting the number of barcodes. Increasing $s_0/\gamma$ results in an increase in the final CFU. Noise strength, $\sigma$, affects the CFU and barcode numbers. A large $\sigma$ can push the population to extinction. Survival probability (bottleneck size), q, affects the number of surviving barcodes and consequently the final total CFU. See Methods for parameter values. Source data are available online for this figure.

probability $1 - q$. This formulation allows us to describe both the intravenous and intratumor experiments with a single parameter, and we expect direct injection into the tumor to correspond to a value of $q$ that is significantly higher than that for intravenous injection. Furthermore, it allows us to determine the initial barcode sizes and the number of surviving barcodes, which we then use to simulate model dynamics.

We infer the value of $q$ directly from the experimental data by predicting the probability of observing a particular barcode in the tumor as a function of its frequency in the inoculum (see Methods). The comparison of this prediction with the data is shown in Fig. 5A,B. To test this prediction over the whole range of possible inoculum frequencies from 0 to 1, we aggregated multiple barcodes at random to achieve higher frequencies in the inoculum and a higher probability of observation in the tumor (Fig. 5A,B). We see that the agreement between theory and experiment is excellent over the full range of frequencies. We find a strong correlation between inoculum frequency and observation probability in the tumor. However, we see no correlation between inoculum and post injection frequencies (Figs. EV2B and EV3B). Moreover, the $q$ values are $2.6 \times 10^{-3}$ and $5.0 \times 10^{-5}$ for the intratumor and intravenous injections, respectively, quantitatively confirming our expectation that direct injection into the tumor significantly widens the bottleneck, in this case by about two orders of magnitude.

The value $q = 2.6 \times 10^{-3}$ allows us to estimate the founding size of barcodes in the intratumor experiments. An inoculum of 5 million cells across roughly 40,000 barcodes (Fig. 3B) corresponds to an average of about 100 cells per barcode. A survival probability of $q = 2.6 \times 10^{-3}$ per cell means that the chance of a 100-member barcode having 0, 1, or 2 cells post-inoculation is 77, 20, or 2.6%, respectively (see Methods). This means that most barcodes go extinct immediately (77%), and the vast majority of the surviving barcodes are founded from a single cell. Because bacterial progeny founded by a single cell are likely to stay proximal within the solid tumor, single-cell founding is consistent with barcodes remaining spatially segregated as they grow. This spatial segregation justifies the heterogeneous resource access among barcodes assumed in the dynamical model above.

### Direct comparison of dynamics and statistics agrees with experimental results

Combining the bottleneck model with Eqs. (3) and (4), we compared our predictions for the dynamics and statistics directly with the experimental data (see Methods for simulation details). Figure 5C,D shows comparisons of the dynamics with the data for the intratumor and intravenous cases, respectively (note in the simulations the immediate drop of both the total CFUs and the barcode number due to the bottleneck, followed by the recovery of the CFUs). Here, we have used the $q$ values inferred above, set $\sigma = \sqrt{g}$, and used plausible values for the maximal growth rate $g = 2\mathrm{hr}^{-1}$ and molecular diffusion timescale $D = 1\mathrm{s}^{-1}$. The final parameter values are set by recognizing that the maximum total CFU number in our model is roughly $N \sim Bs_0/k$, so knowing $N/B$ from experiments allows us to set $s_0/k$. Specifically, we set $s_0/\gamma = 5 \times 10^4$ and use $k/\gamma = 1$ and 0.01 for the intratumor and intravenous cases, respectively. We see in Fig. 5C,D that the dynamics of the model agree very well with the experimental data.

Finally, we used the model to calculate the expected barcode frequencies in the intratumor and intravenous cases. Figure 5E shows the barcode frequencies plotted as a function of barcode rank according to our model and as observed in experiments (similarly to Fig. 3G). The top panels show the plots we generated by the model (using only the $q$ value determined from each of the mouse tumors; see Methods). The bottom panels show the corresponding plots from resected tumors. This comparison shows that simulations for both the intratumor and intravenous cases agree very well with the experimental statistics. Moreover, we observed that while the statistics of the intratumor case follow Zipf's law, the statistics of the intravenous case do not, matching the experimental observations.

## Discussion

Studies of intratumor bacteria are primarily driven by research of the tumor-microbiome found in many types of cancers (Sepich-Poore et al, 2021; Ma et al, 2020; Riquelme et al, 2019) and by research of genetically modified bacteria that are introduced to the tumor niche (Gurbatri et al, 2022; Lu and Tong, 2024). Yet, despite wide interest in intratumor bacteria, basic questions about this phenomenon remain underexplored. Specifically, the population dynamics governing tumor colonization remain largely uncharted. We addressed some of these questions using a murine tumor model, allowing intravenous and intratumor administration of non-pathogenic bacteria. In this model, bacteria exclusively colonize tumors (Fig. EV4). Our controlled experiments revealed that bottlenecks dominate the seeding of colonization and are followed by rapid, yet highly nonuniform growth (Figs. 2B,F and 3B,E). Surprisingly, we revealed that intratumor injections repeatedly gave rise to bacterial progenies that are characterized by a scale-free distribution that matches Zipf's law (Fig. 4E). Our theoretical work then proceeded to explore growth models that can explain the experimental observations.

Our work reveals how the fundamental criteria required for Zipf's law (Saichev et al, 2010) can emerge in microbial population dynamics, as illustrated in Fig. 6A. First, Zipf's law requires the lack of a dynamic attractor (equivalently, a potential minimum), because an attractor would introduce a characteristic scale, yet Zipf's law is scale-free. In our model, no strong attractor exists because the environment is assumed to be very noisy: fluctuations outweigh any deterministic contribution to cell death, which would otherwise balance cell growth to create a characteristic scale for each barcode's population size. Second, Zipf's law requires biased fluctuations, such that lower values of a quantity (here, clone size) are more probable than higher values. In our model, biased fluctuations are provided by multiplicative noise; in fact, the specific choice of multiplicative noise is responsible for the scaling of frequency with inverse rank as opposed to some other power (see Methods). Last, Zipf's law requires a mechanism to prevent extinction. In our model, this mechanism is provided by local growth limitation (the dependence of growth rate on clone size in Eqs. (3) and (4)), which prevents the largest clones from consuming all the resources and driving all smaller clones to extinction. Together, as illustrated in Fig. 6A, these criteria result in a probability distribution whose tail follows a power law with power $-2$, (equivalent to a rank-frequency plot with power $-1$). Figure 6B

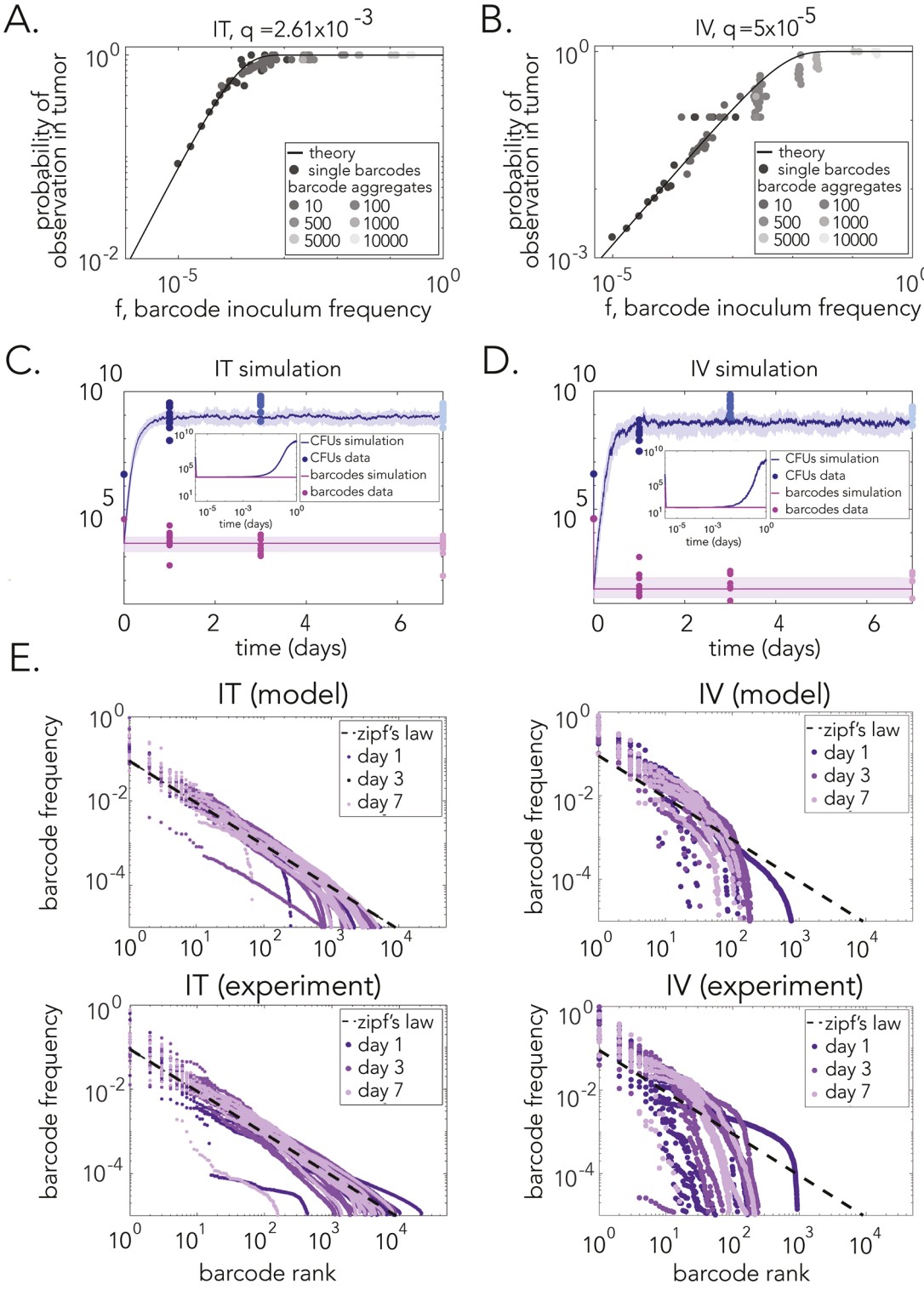

**Figure 5. Mathematical model captures dynamics of bacterial load and barcode frequencies.**

(**A, B**) Bottleneck theory predicts the probability of observing a barcode in the tumor given its inoculum frequency. *q* represents the inferred survival probability from data for the IV and IT injections. Barcode data was aggregated to populate higher frequencies and confirm that they obey the theory. (**C, D**) IT and IV simulations of total CFU and barcode number dynamics. Experimental data is overlaid with circles. Solid dark lines indicate the median value of different stochastic trajectories. Shaded areas represent the range between the lower and upper quartiles. Insets: Immediate post injection dynamics showing the initial bottleneck effect. (**E**) Comparison between IT and IV statistics on different days for both simulations and experiments. IT data satisfies Zipf's law, while IV data does not. See Methods for parameter values. Data information: Data shown from a single experiment. Number of mice $n = 14$ for Panel (**A, E**) (bottom, right) $n = 15$ for panel (**B, E**) (bottom left); Technical replicates $n = 2$. Source data are available online for this figure.

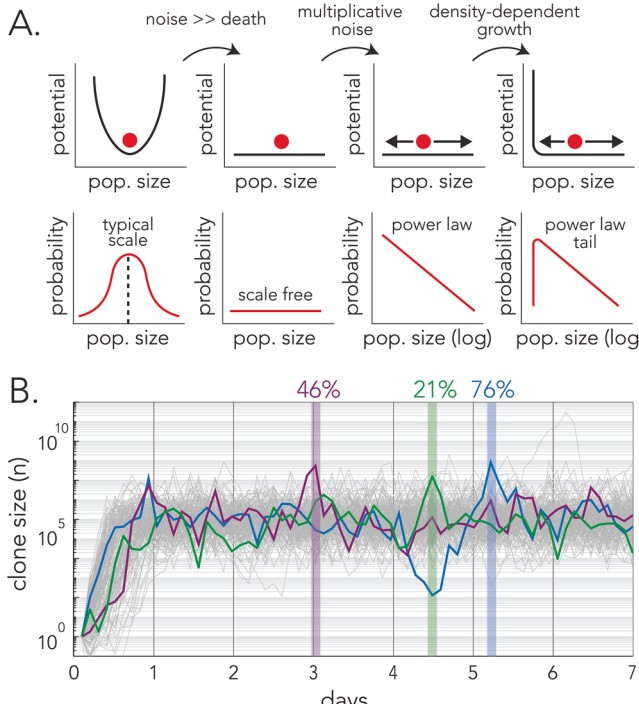

**Figure 6.   The criteria required for Zipf's law emergence and the underlying microbial population dynamics.**

**(A)** Zipf's law requires the lack of a dynamic attractor, biased fluctuations, and extinction prevention. In our model, these criteria are provided by noisy cell death, multiplicative noise, and local growth limitation, respectively. Together, these criteria result in a probability distribution whose tail follows a power law with power $-2$, equivalent to a rank-frequency plot with power $-1$ (Zipf's law). **(B)** The population dynamics expected during tumor colonization after intravenous infection. The graphs show the simulated dynamics in 178 clones successfully colonizing a tumor after intravenous infection. The dynamics of three random clones are highlighted in color, and their respective maximum sizes are shown with vertical lines. The numbers on top mark the proportion of cells from the corresponding same color clone out of all bacteria in the tumor. Source data are available online for this figure.

shows the bacterial population dynamics expected under our colonization model. As the simulation demonstrates, our model predicts that dominant clones will fluctuate over time. Moreover, we expect that at each point in time, most bacteria will belong to only a handful of transient dominant clones.

Zipf's law is also known to arise from mechanisms other than the three criteria above. Mutations can give rise to subpopulations whose sizes can be distributed according to Zipf's law, as in the classic Luria-Delbruck process (Kessler and Levine, 2013) or within spatially expanding populations (Schreck et al, 2023). The clonal dynamics of the adaptive immune system can also be described by geometric Brownian motion, a limit of which is Zipf's law (Mora et al, 2010). These systems are distinct from ours in that new populations arise at various points in time or space due to mutations or migration from a repertoire, whereas in our system, no new barcodes emerge. More broadly, Zipf's law has been argued to arise in many systems as a sign of criticality (Mora and Bialek, 2011; Mora et al, 2010) or due to the presence of unobserved variables (Schwab et al, 2014). It is an interesting open question whether these broader explanations for Zipf's law are applicable to microbial population dynamics. Finally, it

is important to note that multiplicative noise that scales linearly with the population size, which we found is required for Zipf's law, is incompatible with neutral theories of ecology (Maurer and McGill, 2004). In neutral theories, the primary driver is intrinsic stochasticity (Maurer and McGill, 2004; Fisher and Mehta, 2014), which has a sublinear scaling with population size (Gillespie, 2000). Therefore, neutral theories of ecology cannot explain the scaling of Zipf's law that we observe here.

The work presented here relies on a well-established yet simple animal model of bacteria colonized tumors (Geller et al, 2017; Yu et al, 2019; Leventhal et al, 2020; Gentschev et al, 2022; Gurbatri et al, 2024; Wu et al, 2023). However, the combined experimental and theoretical framework we developed can serve as the basis for two important future research directions. The first direction is continued exploration of principles underlying tumor colonization. This line of work will use highly-controlled experiments to systematically map and disentangle how discrete parameters of tumor and bacterial biology impact the tumor-microbiome. These experiments can include more complex animal models, such as orthotopic implantations or genetically engineered mouse models, which better reflect the native tumor environment, and even patient-derived xenografts that more closely mimic human tumor heterogeneity. These host models can reveal if bacterial population dynamics are altered by the tumor conditions. Additional experiments can be designed to explore key parameters of the colonizers and can include multiple bacterial infections, multi-species colonization, and the impact of different selection pressures, such as those imposed by antibiotics. While the study of intratumor selection pressures by administrating a pooled collection of mutated strains is impractical given the population dynamics we discovered (bottlenecks and uneven growth), it might be still feasible by inducible transposon mutagenesis (Basta et al, 2025) that will be triggered only after colonization.

The second direction to which our work contributes is the ongoing discussion in the field about the origin of bacteria in tumor samples. While recent publications argued against a few studies in the field (Gihawi et al, 2023; Poore et al, 2020, 2024; Sepich-Poore et al, 2024; Fletcher et al, 2023; Aykut et al, 2019), a broader concern is whether evidence of intratumor bacteria arises from PCR amplification of bacterial contaminants, as was previously determined for placenta and brain tissues. Our work suggests that descriptive statistics of bacterial populations can provide strong evidence for a genuine tumor-microbiome. Specifically, if frequencies of same-species clones across a tumor follow Zipf's law, they very likely arose from the unique growth dynamics characteristic of the tumor environment. While our approach relied on genetic barcodes, alternative terminal assays can also be performed on resected tumor tissues. Previous works have applied various staining approaches for detecting bacteria in tumor sections and have often detected highly nonuniform localization, with bacterial signal frequently forming separated foci (Dang et al, 2025; Geller et al, 2017; Nejman et al, 2020). Specifically, if human tumors are initially seeded by a low number of bacteria, as we observed in our mouse models, and clones arise from spatially separated single cells, the spatial bacterial signal should follow similar statistics. Therefore, species-specific bacterial staining of tumor sections (Niño et al, 2022) may allow distinguishing genuine tumor microbiomes from contamination.

# Methods

### Reagent and tools table

| Reagent/resource | Reference or source | Identifier or catalog number |
|---|---|---|
| **Experimental models** | | |
| 5-week-old female BALBc/J mouse (*Mus musculus*) | Jackson Lab | Stock Number:000651 |
| *Escherichia coli* Nissle 1917 (EcN) | ArdeyPharm GmbH (Pharma Zentrale GmbH), Germany | NA |
| EcN Barcoded Library | This study | NA |
| CT-26 Cell Line | Straussman Lab | RRID:CVCL_7256 |
| CT-26 Cell Line with histone H2B-mCherry | This study | Derived from RRID:CVCL_7256 |
| **Recombinant DNA** | | |
| Kanamycin resistance cassette | Derived from the gDNA of a single strain from the Dharmacon KEIO knockout collection | Horizon Discovery Cat# OEC4987, OEC4988 |
| pSIM6 plasmid | https://ncifrederick.cancer.gov/recombineering/strains-plasmids-and-primers | NA |
| Plasmid pGEN-luxCDABE | Addgene (Lane et al Proc Natl Acad Sci USA. 2007 Oct 16;104(42):16669-74. Epub 2007 Oct 9.) | Addgene Plasmid #44918 |
| H2B-mCherry and Lentivirus carrying H2B-mCherry construct | Shraga, A. et al, Covalent docking identifies a potent and selective MKK7 inhibitor. Cell Chem. Biol. 26, 98–108 (2019). | Cloned and prepared for this study using the constructs from the cited source |
| **Antibodies** | | |
| NA | | |
| **Oligonucleotides and other sequence-based reagents** | | |
| Target_lacZ_F Primer | This study | 5'GTTGTGTGAAATTATGAGCGGATAACAATTTCACACAGGATACAGCTATTCCGGGGATCCGTCGACC3 |
| Target_lacZ_barcode_R Primer | This study | 5'ACGGGCAGACATAGCCTGCCCGGTTATTATTATTTTTGACACCAGACCAANNNNNNNNNNNNNN NNNNNNNNTGTAGGCTGGAGCTGCTTCG 3' |
| Amplicon_Fwd Primer | This study | 5' TCGTCGGCAGCGTCAGATGTGTATAAGAGACAG(1–3 N) cctgcccggttattattattttttg 3' |
| Amplicon_Rev Primer | This study | '5 GTCTCGTGGGCTCGGAGATGTGTATAAGAGACAGgattcatcgactgtggcc 3' |
| **Chemicals, enzymes and other reagents** | | |
| Lysogeny Broth (LB) with or without agar | https://cshprotocols.cshlp.org/content/2016/9/pdb.rec090928.full?sid=f82eea5d-9c8b-496b-ac4e-20b8dc3c0aec | |
| M9 minimal media | https://cshprotocols.cshlp.org/content/2010/8/pdb.rec12295, Carbon source was replaced with glycerol or sodium acetate for experimental conditions | |

| Reagent/resource | Reference or source | Identifier or catalog number |
|---|---|---|
| SOC Media | https://cshprotocols.cshlp.org/content/2018/3/pdb.rec098863.full?sid=0cc96c84-6181-47a5-9e1b-99fe31182b40 | |
| Tryptone | Fisher Scientific | Cat# BP1421-500 |
| Yeast extract | THERMO - DIFCO BACTO | Cat# 212720 |
| Sodium chloride (NaCl) | Fisher Scientific | Cat# FLS2713 |
| Agar | Fisher Bioreagents | Cat# BP2641-500 |
| Kanamycin sulfate | Thermo Scientific Chemicals | Cat# 611290050 |
| M9 minimal salts | BD | Cat# 248510 |
| Magnesium sulfate (MgSO$_4$) | Fisher Chemicals | Cat# M65-500 |
| Potassium chloride (KCl) | Sigma-Aldrich | Cat# P3911-500G |
| Protein hydrolysate amicase | Sigma-Aldrich | Cat# 82514-1KG |
| Dextrose (D-glucose), anhydrous | Fisher Scientific | Cat# D16-1 |
| Glycerol | Fisher Scientific | Cat# BP229-1 |
| Sodium acetate | Sigma-Aldrich | Cat# S2889-250G |
| RPMI 1640 media with L-glutamine | Gibco | Cat# 11875093 |
| Fetal bovine serum | Gibco | Cat# 26-140-079 |
| Hygromycin B | Corning | Cat# 30-240-CR |
| TripLE™ Express Enzyme | Gibco | Cat# 12605010 |
| Phosphate-buffered saline (PBS) | Corning | Cat# 21-040-CV |
| Q5 DNA polymerase | New England Biolabs | Cat# M0491L |
| Agarose | Fisher Scientific | Cat# BP160500 |
| 10X TRIS-ACETATE-EDTA TAE Buffer | Fisher Scientific | Cat# BP13354 |
| Disposable scalpels | Exel International | Cat# 29550 |
| Air-Tite Sterile Syringe with Needle 1cc Luer Slip, 27 G ½" | Air-Tite | Cat# MS12712 |
| Insulin Syringe, 1cc, 28G1/2 | Exel Scientific | Cat# 26027 |
| Lyzing matrix I tubes | MP Biomedicals | Cat# 6918100 |
| Zymo Quick-DNA Midiprep Plus Kit | Zymo Research | Cat# 4075 |
| Qubit High-Sensitivity DNA Reagent | Thermo-fisher | Cat#Q32854 |
| Quant-iT 1X dsDNA HS Assay | Thermo-fisher | Cat#Q33232 |
| 2x KAPA HiFi HotStart Ready Mix | Kapa Biosystems | Cat#KK2602 |
| Illumina Nextera XT Index Kit v2 Sets A/B/C/D | Illumina | Cat# FC-131-2001 FC-131-2002 FC-131-2003 FC-131-2004 |
| ZR-96 Zymoclean Gel DNA Recovery Kit | Zymo Research | Cat#: D4022 |
| Nextseq 500/550 High Output Reagent Kit, 75-cycles | Illumina | Cat# 20024906 |
| Zymo Bacterial/Fungal DNA extraction kit | Zymo Research | Cat# D6005 |
| **Software** | | |

| Reagent/resource | Reference or source | Identifier or catalog number |
|---|---|---|
| Bartender | Zhao et al, 2017 | https://github.com/LaoZZZZZ/bartender-1.1 |
| Matlab | Mathworks | https://www.mathworks.com/products/matlab.html |
| shadedErrorBar Function for MATLAB | Rob Campbell (2025). raacampbell/shadedErrorBar (https://github.com/raacampbell/shadedErrorBar), GitHub. Retrieved October 6, 2025. | |
| Adobe Illustrator | Adobe | https://www.adobe.com/ |
| Living Image® for IVIS Spectrum CT | Perkin-Elmer / Revvity | https://www.revvity.com/, https://www.perkinelmer.com/ |
| **Other** | | |
| Fastprep-24 instrument | MP Biomedicals | Cat# 116004500 |
| Illumina NextSeq 550 | Illumina | https://www.illumina.com/systems/sequencing-platforms/nextseq.html |
| Illumina Whole-Genome Sequencing Service, 400 Mbp | SeqCenter | https://www.seqcenter.com/ |
| Eon Plate Reader | Agilent (BioTek) | https://www.agilent.com |
| Tecan Spark Plate Reader | Tecan | https://www.tecan.com/spark-overview |
| IVIS Spectrum CT | Perkin-Elmer | https://www.perkinelmer.com/ |

## Bacteria and growth conditions

*E. coli* Nissle 1917 strain (Ardeypharm, GmbH, Germany) was used for cloning the barcoded strain collection. For all in vivo experiments, bacteria were inoculated into Lysogeny broth (LB) containing 50 μg/mL kanamycin and grown overnight at 37 °C, 200 rpm orbital shaking. In the in vitro experiment, it was performed in four different media types that were supplemented with 50 μg/mL kanamycin. We used LB for nutrient-rich experiments and M9 for nutrient-poor experiments (M9 minimal media with 0.2% amicase). Nutrient-poor media was supplemented with one of these carbon sources: 0.4% glucose, 0.4% glycerol, or 0.4% acetate.

## CT-26 cell line culture and generation of CT-26 with Histone H2B-mCherry tag

CT-26 cells were cultured in RPMI media containing 5% fetal bovine serum, at a 37 °C incubator with 5% $CO_2$. For subcutaneous injection, cells were collected from the plates using TripLE, washed, and resuspended in PBS at $1 \times 10^7$ cells/mL. To prepare CT-26 cells with an H2B-mCherry tag, CT-26 cells were infected with a Lentivirus carrying an H2B-mCherry sequence. Cells with H2B-mCherry were selected by adding 300 ug/mL hygromycin to the growth media.

## Cloning the barcoded library

Kanamycin resistance cassette with 20 random nucleotide barcodes, targeting *lacZ* locus of *E. coli* Nissle 1917 genome, was amplified using the following forward and reverse primers:

Target_lacZ_F:5'GTTGTGTGAAATTATGAGCGGATAACAATT TCACACAGGATACAGCTATTCCGGGGATCCGTCGACC3' and Target_lacZ_barcode_R:5'ACGGGCAGACATAGCCTGCCC GGTTATTATTATTTTTGACACCAGACCAANNNNNNNNNN NNN NNNNNNNTGTAGGCTGGAGCTGCTTCG 3'. We used the genomic DNA of a knockout strain from the KEIO collection as a template, amplifying the kanamycin resistance cassette (Baba et al, 2006). The PCR product was purified from an agarose gel. Competent cells with induction of beta, gam, and exo genes from the pSIM6 plasmid (Datta et al, 2006) were prepared from a single transformant as described previously (Juhas and Ajioka, 2016). We transformed 2.5 ug of the PCR product into aliquots of competent cell and plated them on selective media agar plates. We collected roughly 50,000 single colonies from multiple plates after overnight incubation at 37 °C. The colonies were scraped from the agar plates and homogenized in PBS before freezing the collection by adding 40% glycerol (1:1 v/v) and storing it at −80 °C. To cure the pSIM6 plasmid, the pooled library was inoculated and grown for three days in liquid LB containing kanamycin, passaging daily (1:2000 dilution). The final library was stored as glycerol stock (20% glycerol v/v) at −80 °C.

## Barcoded library injection

We thawed 100 μL of the barcoded library glycerol stock and resuspended it in 20 mL of LB supplemented with kanamycin and grew the culture overnight. The next morning, we diluted the culture 1:100 into 30 mL of LB supplemented with kanamycin and grew the culture until reaching an $OD_{600}$ of 0.5. We pelleted and washed the bacteria with PBS three times. Washed bacteria were incubated at room temperature for an hour, and the $OD_{600}$ was measured. Considering $OD_{600}$ of 1 culture contains $4.71 \times 10^8$ CFU/mL. For i.v. injections, we prepared $5 \times 10^7$ CFU/mL. For i.t. condition, we prepared $2.5 \times 10^8$ CFU/mL. Four technical replicates of timepoint zero samples were frozen as glycerol stock, and gDNA was isolated directly from these stocks.

## Mouse experiments

### Ethics statement
All of the mouse experiments in this manuscript have been approved by the University of Massachusetts Chan Medical School, Institutional Animal Care and Use Committee (IACUC), with the protocol number PROTO202100199.

### Number of animals requested
As a preliminary experiment, it was not possible to use power analysis to estimate the required population size. A previous bottleneck experiment using a systemic infection model in mice used nine to ten animals per group (Hullahalli and Waldor, 2021). By using a two-tumor mouse model, we chose to use half the number of animals per condition, thus requesting five per group. A total of three experimental repeats was requested to allow us to evaluate inter-experiment variation (if needed).

### Housing and husbandry conditions of animals
Animals were housed in five mice per cage with access to food and water. Cages were cleaned on a weekly schedule.

### Health monitoring

Starting from cancer cell injection, the health of the mice was monitored every two to four days based on the following criteria: body weight, checking palpable tumor and measuring the tumor dimensions, activity, fur condition, posture, tumor ulceration and body condition score. A veterinarian was consulted for any animal that does not meet the minimum health criteria set in the IACUC protocol and UMass Chan Medical School policies, and the mouse would be euthanized regardless of the planned experimental endpoint.

### Randomization

Assignment of animals to treatment groups (i.v. or i.t. injections of bacteria) was random. However, in order to minimize number of animals used in the study (and avoid unnecessary repeats), mice with smaller tumor sizes were allocated to later euthanasia timepoints (Day 3/Day 7) within each treatment group. This avoided loss of biological replicates during the experiment due to outgrowth of the tumor and the requirement for euthanasia for a humane endpoint.

### Blinding

No blinding was done.

## Mouse experiment in the main figures: tumor colonization with barcoded library

Thirty 5-week-old female BALB/cJ mice (Jackson Laboratory, Bar Harbor, Maine) were allowed to acclimate in the animal facility for 7 days. To form subcutaneous tumors, all mice were injected with $1 \times 10^6$ CT-26 cells on both flanks subcutaneously and tumor growth was monitored every 2–4 days. When tumors reached a volume of 300–500 mm³, mice were injected with barcoded bacteria in two groups (15/14 mice each group). Mice in the *i.t.* group were injected with $5 \times 10^6$ CFU per tumor, and mice in the *i.v.* group were injected with $10^7$ CFU per mouse by tail vein injection. Mice were then euthanized according to experimental endpoints on days 1, 3, or 7 after bacteria injection and tumors were flash-frozen and kept at −80 °C for later analysis. Exceptions to tumor development: One mouse in the *i.v.* group (day 1) did not develop a tumor on the left flank and had a single tumor. One mouse, to be assigned to *i.v.* (day 7), did not develop any tumors and was excluded from the study.

## Mouse experiment in Fig. EV4: tumor colonization with luminescent bacteria

Ten 5-week-old female BALB/cJ mice (Jackson Laboratory, Bar Harbor, Maine) were allowed to acclimate in the animal facility for 7 days. To form subcutaneous tumors, all mice were injected with $1 \times 10^6$ CT-26 cells tagged with histone H2B-mcherry fluorescence on both flanks subcutaneously and tumor growth was monitored every 2–4 days. When tumors reached a volume of 300–500 mm³, mice were injected with $1 \times 10^7$ CFUs of *EcN* from the tail vein. To monitor bacterial localization, mice were imaged by IVIS Spectrum CT on days 1, 3, 5, and 7 post bacteria injection. Mice were anesthetized by placing them in a chamber with 2.5% isoflurane. Then mice were moved into the IVIS Spectrum CT, in which isofluorane administration was continued by nose cones.

Luminescent images were captured with a 1-min exposure time, and the stage was kept at 37 °C. Mice were then returned to their cages and monitored for recovery.

Mice were euthanized following the last imaging on day 7. Mouse 8 was excluded from the study since it developed an ulcerated tumor and has been euthanized before reaching the imaging timepoints.

## Justification for cell lines

The CT-26 cell line used in this article was tested as Mycoplasma-free, and the cell identity was confirmed by STR profiling (ATCC). STR profile was 98% similar to CT-26.CL25 (ATCC, #CRL-2639).

## Determining bacterial CFU in tumors

Tumors removed from euthanized mice were finely cut with a sterile scalpel, and 200 mg of processed tumor was added to lyzing matrix I tubes filled with 300 μL PBS. Tumor samples were homogenized twice at a speed of 6 m/s for 40 s using the MP Biomedicals Fastprep-24 instrument. 20 μL of the homogenate was used to make serial dilutions of 1:1000, 1:10,000 and 1:100,000 in 200 μL of PBS. About 100 μL of these serial dilutions were plated on selective LB plates containing kanamycin. Plates were incubated overnight, and the next day, colonies were counted. We determined the CFU by counting colonies on dilution plates that had 100–500 colonies. The weights of homogenized samples and the total tumor weights were used to calculate the CFU per gram of tumor, and the total CFU per tumor. For some tumor samples, we repeated the CFU counting once again after freezing the homogenized tumor. In these cases, we used the average CFU numbers from frozen and fresh tumors for downstream analysis.

## Sequencing library preparation

We purified DNA from the homogenized frozen tumor samples using the Zymo Quick-DNA Midiprep plus kit and determined the DNA concentration using Qubit High-Sensitivity DNA reagent. Since the total gDNA contains a high amount of mouse DNA relative to the bacterial DNA, we used 10 μg of DNA as a template for each PCR. A region of ~400 bp around the barcoded region was amplified using 2x KAPA HiFi HotStart Ready Mix and the following forward and reverse primers: Amplicon_Fwd Primer: 5' TCGTCGGCAGCGTCAGATGTGTATAAGAGACAG(1–3 N)cctg cccggttattattattttttg 3' and Amplicon_Rev Primer: '5 GTCTC GTGGGCTCGGAGATGTGTATAAGAGACAGgattcatcgactgtggcc 3'. We set 4-7 PCR reactions were per tumor sample, depending on the amount of DNA extracted from the tumor tissue. We pooled together 10 μL of each of these technical replicate PCR and ran the mixture on a 3% agarose gel. We then purified the amplicon from the gel and determined the DNA concentration using Qubit high-sensitivity DNA reagent. A second 13-cycle PCR was performed using Illumina Nextera XT indexes and 2x KAPA HiFi HotStart ReadyMix for multiplexing. The products were run on 3% agarose gel and purified again. Libraries were normalized to the same concentration, denatured, and diluted according to Illumina NextSeq System Denature and Dilute Libraries Guide. Sequencing was performed using Nextseq 500/550 High Output Reagent Kit, 75-cycles on Illumina NextSeq 500 device. We

sequenced each tumor sample as two technical replicates to validate that our library preparation procedure did not introduce any bottlenecks in barcode amplification before deep amplicon sequencing.

## Targeted barcode sequencing analysis

We used Bartender (Zhao et al, 2017) to extract and cluster the barcodes from raw fastq files. We first ran bartender extractor (bartender_extractor_com) with the following parameters: Average base quality score cutoff of 30 and allowing no mismatch on upstream and downstream anchor sequences: "-q ? -p ACCAA[20] TGTAG -m 0". Next, we ran Bartender Clustering (bartender_single_com) with the following parameters: "-d 2 -z 10 -l 5 -t 1 -s 1". A MATLAB script was used to further organize the barcode clusters. First, we generated a master list of barcodes that was based on the four technical T0 samples. This master barcode list defines the identification and the total number of barcodes present in the collection. The list of T0 barcodes was used to filter out any new barcodes that were observed only in mouse tumors (the vast majority of excluded barcodes originate from errors in DNA replication, PCR, and next-generation sequencing). For a barcode to appear in the master barcode list, it had to be present in two out of four T0 replicates with a normalized frequency above 10 RPM (39,091 barcodes in total). Since each tumor sample was processed and sequenced as two technical replicates, an average RPM value was calculated.

## Whole-genome sequencing of bacteria isolated from tumors

We isolated individual clones directly from tumors by plating aliquots of tumor tissue homogenate on selective LB agar plates. We then streaked three individual colonies from each of the plates on a new selective LB agar plate. We, Sanger sequenced the barcode region of each pure cultures after a colony PCR amplification and tested if its barcode matches the most frequent barcode of the matching tumor (the identity of the dominant barcode was determined by deep amplicon sequencing of the tumor sample). After identifying isolates matching the dominant barcode from ten individual tumors, we extracted DNA from each of these clones and sent samples (10 total) for whole-genome sequencing at SeqCenter (Illumina paired-end sequencing with $2 \times 151$ bp, 400 Mbp yield). The average coverage per genome was x230. We double-checked that the barcode found in genomic sequences matches the respective dominant barcode of the matching tumor. Reads were aligned to the reference genome (NCBI CP058217) using breseq (Deatherage and Barrick 2014) to identify mutations. Called mutations were compared to each other to identify if any clone-specific mutations exist.

## In vitro growth experiments

We thawed 15 μL of the barcoded library glycerol stock and added it to 6 mL each of four media types used. We grew cultures overnight at 37 °C in a 200 rpm shaker. The next day, $OD_{600}$ of the cultures was measured, and each culture was diluted to an $OD_{600}$ of 0.5 before further diluting them 1:200 into 80 mL of their respective media type. Each culture condition was plated into three 96-well plates for growth (each 96-well plate was considered a technical replicate at this point). Cultures were incubated on a benchtop incubator at 37 °C at 750 rpm shaking. We simultaneously monitored the growth of identical cultures grown in a single 96-well plate with a plate absorbance reader growing in similar conditions (Biotek/Eon). We collected samples from all technical replicates of a specific growth condition when absorbance reached ~0.4 in the plate reader. Cultures from each of the 96-well plates were pooled together and centrifuged at 4 °C to pellet the cells. Genomic DNA was isolated from the cell pellets using the Zymo Bacterial/Fungal DNA extraction kit. Extracted gDNAs were normalized to 40 ng/μL, and a total of 500 ng input was used for the NGS library preparation. Similar library preparation and sequencing approach as previously described for mouse tumor samples was followed, however since these were pure bacteria cultures only single reaction per sample was set-up. Sequencing data was analyzed using the same parameters as the mouse experiments.

## Effects of model parameters on the dynamics

We see in Fig. 4D that the final total CFU count, but not the final barcode number, is primarily determined by the ratio $s_0/\gamma$. This makes sense because the background nutrient concentration $s_0$, converted to cells via the yield coefficient $\gamma$, ultimately determines the total number of cells that can be sustained. We see in Fig. 4E that the probability that cells go extinct (as well as the amount of fluctuations) is primarily determined by the noise strength $\sigma$. This makes sense because the noise term, when negative, represents an effective cell death in our model. Specifically, when it overpowers the growth term ($\sigma \gg \sqrt{g}$; in Fig. 4E $g = 2hr^{-1}$), total extinction is very likely. Finally, we see in Fig. 4F that the final number of surviving barcodes (and to some degree the final total CFU count) is primarily determined by the bottleneck size. This makes sense because shrinking the bottleneck reduces the probability that any cell survives the inoculation process, and barcodes with no surviving cells post-inoculation are not present at later times. Because this is a continuum model, we choose an arbitrary threshold for barcode extinction (set to $N_i < 1$ in Fig. 4). We later relax this condition in Fig. 5, as barcode extinction only occurs immediately after injection when the number of cells per barcode is small. Consequently, the inferred bottleneck values from the experiments account for all initial extinction sources.

## Theoretical derivation of Zipf's law from the stochastic model

To find the statistics from the stochastic model in Eqs. (3) and (4), we first utilize the fact that the statistics are measured after growth has significantly slowed down. At this point, $kN_i \gg s$, and Eqs. (3) and (4) simplify to

$$\frac{dN_i}{dt} = \frac{gs}{k} + \sigma\eta_i N_i \tag{5}$$

$$\frac{ds}{dt} = -\gamma B \frac{gs}{k} + D(s_0 - s) \tag{6}$$

where $\sum_{i=1}^{B} 1 = B$. Equation (6) is solved at steady state and substituted into Eq. (5) to give

$$\frac{dN}{dt} = \frac{gDs_0}{Dk + Bg\gamma} + \sigma\eta N \qquad (7)$$

The index ($i$) is dropped since the dynamics of all barcodes is equivalent. Now, we are left with a simple one-dimensional Langevin equation with a constant force term and a noise term.

Equation (7) is solved using a Fokker-Planck equation, $\partial p(n,t)/\partial t = -\partial/\partial n[\mu(n,t)p(n,t)] + \partial^2/\partial n^2[\sigma(n,t)p(n,t)]$, where $\mu(n,t) = gDs_0/(Dk + Bg\gamma)$, and $\sigma(n,t) = \sigma^2 n^2/2$. Therefore, the dynamics of $p(n,t)$ is given by

$$\frac{\partial p(n,t)}{\partial t} = -\frac{gDs_0}{Dk + Bg\gamma}\frac{\partial p(n,t)}{\partial n} + \frac{\sigma^2}{2}\frac{\partial^2}{\partial n^2}\left[n^2 p(n,t)\right] \qquad (8)$$

Solving Eq. (8) at equilibrium gives

$$p(n) = \frac{n_*}{n^2} e^{-n_*/n} \qquad (9)$$

where $n_* = 2gDs_0/(Dk + Bg\gamma)\sigma^2$. Then, we integrate Eq. (9) to find the complement to the cumulative distribution

$$c(n) = \int_n^\infty p(x)dx$$
$$c(n) = 1 - e^{-n_*/n} \qquad (10)$$

In general, $c(n)$, is inversely proportional to the rank, $r$. This can be seen by considering the effect of different abundances, $n$, on $c(n)$ and $r$. As $n$ increases, $c(n)$ decreases and a barcode with $N_i = n$ is assigned a smaller $r$. The proportionality relation is $c(n) = r/B$. The frequency-abundance relation is straightforwardly given by $f = n/N$, where $N = \sum_{i=1}^{B} N_i$. Consequently, the stochastic model gives the rank-frequency relation

$$f = \frac{-n_*}{N \ln(1 - r/B)} \qquad (11)$$

For high abundances, Eq. (11), to leading order, becomes $f \sim r^{-1}$ (Zipf's law). Equation (11) predicts that Zipf's law is lost at lower abundances (large $r$) and for populations with fewer surviving barcodes (small $B$). This is consistent with intravenous injection not satisfying Zipf's law, where the number of surviving barcodes is much smaller than intratumor injection.

## Bottleneck model

A bottleneck is modeled as a survival probability, $q$, for individual cells. $q = 0$ represents the full extinction of all barcodes before reaching the tumor, while $q = 1$ means that all cells successfully reach the tumor. In other words, each cell is a Bernoulli trial with a probability of success, $q$. Thus, the probability of $x$ cells successfully reaching the tumor for a barcode of $N_i$ cells and inoculum frequency $f_i$ is given by the binomial distribution $g(x) = \binom{N_i}{x} q^x (1 - q)^{N_i - x}$, and substituting $N_i = Nf_i$, the distribution becomes

$$g(x) = \binom{Nf_i}{x} q^x (1 - q)^{Nf_i - x} \qquad (12)$$

Note that with $Nf_i = 100$ and $q = 2.6 \times 10^{-3}$, we obtain $g(0) = 77\%$, $g(1) = 20\%$, and $g(2) = 2.6\%$, as stated in the main text. The probability of observing a barcode in the tumor is the probability that any number of cells with this barcode survive. Hence, $g_{obs}(f) = 1 - g(0) = 1 - (1 - q)^{Nf_i}$. We calculate the predicted mean of surviving barcodes from theory as a function of $q$ and $f_i$

$$\overline{B} = \sum_i^B 1 - (1 - q)^{Nf_i} \qquad (13)$$

where $f_i$ is the inoculum frequency of barcode $i$, and $f_i$ and $\overline{B}$ are both known from data. $q$ is straightforwardly found numerically for intratumor and intravenous injections by plotting both sides of Eq. (13) and finding the $q$ value at the intersection of both curves. The inferred $q$ value is likely an underestimation because it encompasses additional effects from the dynamics beyond the initial bottleneck, such as barcode death due to stochasticity in the tumor environment.

## Simulation

To determine the appropriate bottleneck size, $q$, for capturing the correct range of surviving barcodes, $B$, in the experiment, we simulated the dynamics using several $q$ values that give different $B$ values by day 1. Then, we fitted the $q$ and the resulting $B$ values to a nonlinear function in the form $B = aq^b$. Using the fitted function, we invert the relation to infer the appropriate $q$ values from the experimental $B$ values. The same inference procedure was applied to intratumor and intravenous injections, separately.

After determining the appropriate $q$ values for intratumor and intravenous injections, as explained in the paper, these values are used to simulate intratumor and intravenous injections. First, barcode sizes are initialized using the inoculum dataset used in the experiments to ensure similar initial conditions between theory and experiment. Then, the bottleneck is applied to the inoculum data using Eq. (12). Right after injection, the population goes through a significant decrease in population size as well as in the number of surviving barcodes. This gives the initial population that successfully reaches the tumor. Afterwards, we use Eqs. (3) and (4) to simulate the dynamics. We discretize Eqs. (3) and (4) using Euler's method. We repeat the same procedure for different days for intratumor and intravenous, such that we have a simulation curve corresponding to each experimental curve with the same final $B$ and a unique $q$ for this specific day and injection method. This provides a direct comparison between theory and experimental curves across days and conditions (Fig. 5E).

In all simulations, the noise term, $\eta$, is modeled as a delta-correlated noise with a mean of 0 and a standard deviation of 1. Except for $q$ and $k$, all parameters are kept the same for intratumor and intravenous simulations. $k$ is set to 1 and 0.01 for intratumor and intravenous simulations, respectively, to reflect the ecological niche of each barcode. $k$ is smaller for intravenous, which allows for a larger ecological niche per barcode, resulting from having fewer surviving barcodes compared to intratumor. Other parameters are set to $g = 2hr^{-1}$, $D = 1s^{-1} = 3600hr^{-1}$, $\sigma = \sqrt{g}$,

and $s_0/\gamma = 50000$ which is set to match the final total population CFU.

### In vitro limit

In the in vitro experiment, the population was not subject to an initial bottleneck. Therefore, local niches do not form, and different barcode populations remain well-mixed. In this case, the growth function becomes the Monod growth function. Another important distinction between tumor and in vitro growth dynamics is the source of noise in each. In the tumor, external noise is the dominant form, whereas in the in vitro experiment, intrinsic noise—due to the randomness in the division time of individual cells—dominates. The final in vitro model is given by

$$\frac{dN_i}{dt} = \frac{gs}{s+k}N_i + \eta_i \sqrt{\frac{gs}{s+k}N_i} \tag{14}$$

$$\frac{ds}{dt} = -\gamma \sum_{i=1}^{B} \frac{gs}{s+k}N_i + D(s_0 - s) \tag{15}$$

Comparison between simulations of Eqs. (14) and (15) and in vitro experimental results is illustrated in Fig. EV1. The simulation uses a free parameter to account for the initial number of viable cells that can grow and divide out of the overnight stationary culture.

## Data availability

Barcode Sequencing data have been deposited in NCBI SRA under bioproject IDs PRJNA1136987 and PRJNA1136830. Whole-genome sequencing data of tumor isolated colonies have been deposited in NCBI SRA under bioproject PRJNA1321117. All of the custom scripts developed for the bioinformatics analysis and generating the figures in this paper have been deposited on GitHub (https://github.com/Mitchell-SysBio/Tumor-Colonization-2025.git).

The source data of this paper are collected in the following database record: biostudies:S-SCDT-10_1038-S44320-025-00175-5.

## Peer review information

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

## Acknowledgements

A. Mitchell and S. Sayin were supported by a grant from the National Institute of General Medical Sciences (NIGMS) R35GM133775. A. Mugler and M. ElGamel were supported by NSF award number DMS-2245816 and NIGMS award number R35GM156451. M. ElGamel was supported by the Andrew Mellon Predoctoral Fellowship from the University of Pittsburgh. This research was supported in part by the University of Pittsburgh Center for Research Computing, RRID:SCR_022735, through the resources provided. Specifically, this work used the H2P cluster, which is supported by NSF award number OAC-2117681.

## Author contributions

**Serkan Sayin**: Conceptualization; Data curation; Formal analysis; Investigation; Methodology; Writing—original draft; Writing—review and editing. **Motasem ElGamel**: Formal analysis; Investigation; Methodology; Writing—original draft; Writing—review and editing. **Brittany Rosener**: Investigation; Methodology; Writing—review and editing. **Michael Brehm**: Supervision; Methodology. **Andrew Mugler**: Formal analysis; Supervision; Funding acquisition; Investigation; Methodology; Writing—original draft; Writing—review and editing. **Amir Mitchell**: Conceptualization; Formal analysis; Supervision; Funding acquisition; Investigation; Methodology; Writing—original draft; Writing—review and editing.

Source data underlying figure panels in this paper may have individual authorship assigned. Where available, figure panel/source data authorship is listed in the following database record: biostudies:S-SCDT-10_1038-S44320-025-00175-5.

## Disclosure and competing interests statement

The authors declare no competing interests. Author Serkan Sayin is currently employed by Caris Life Sciences.

# Expanded View Figures

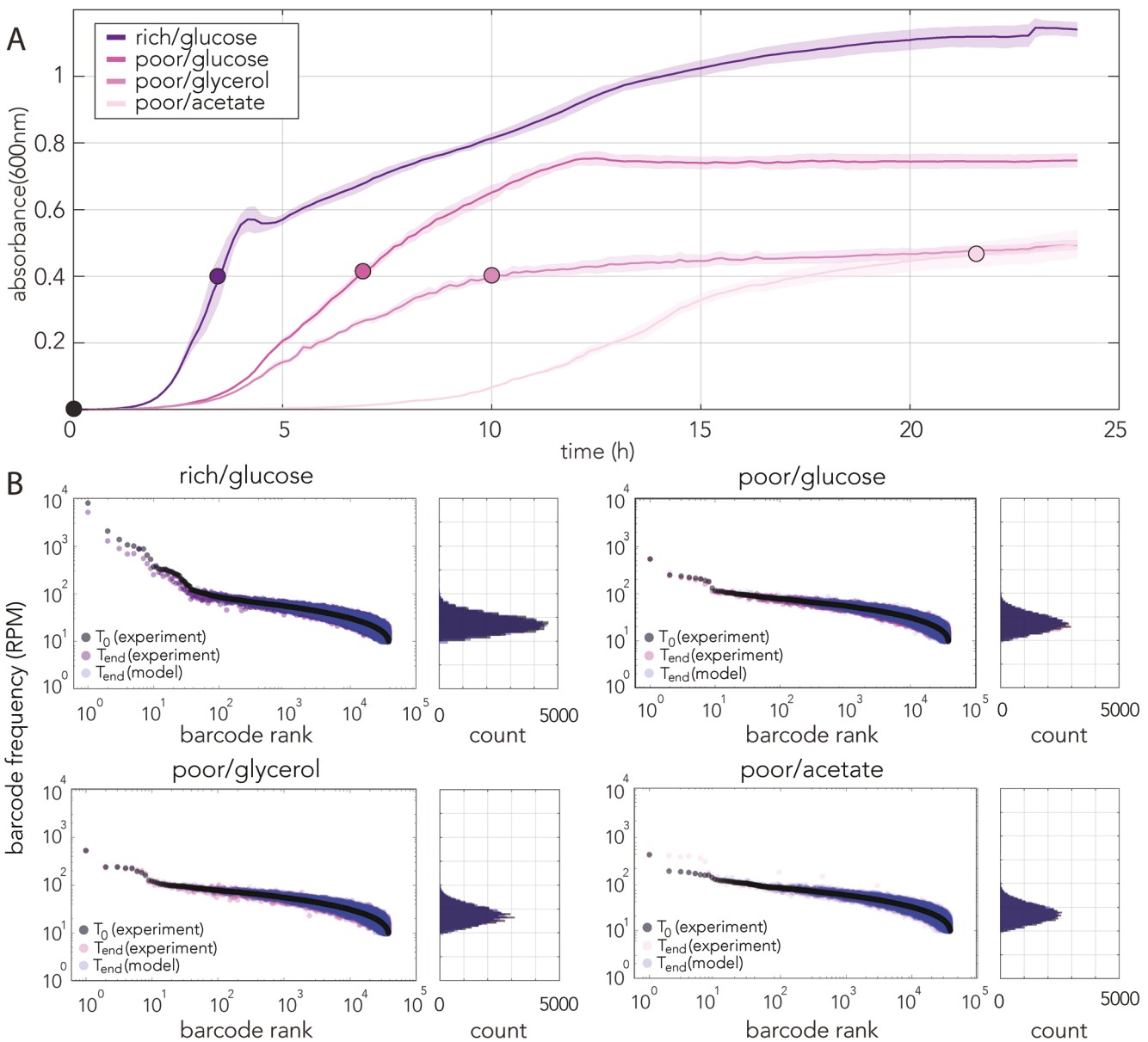

**Figure EV1.  Variation in barcode frequency after in vitro growth follows the Monod model with intrinsic noise.**

(**A**) Growth of barcoded strain collection and isolation points for barcode extraction. Error shades represent standard deviation. (**B**) Variation in barcode frequency before and after growth. The frequency of barcodes in the inoculum is shown and compared to experiment and model simulation results without reranking to show growth noise (see Methods for in vitro model). Intrinsic noise does not change the relative frequency of barcodes; reranking results in the same rank-frequency distribution as the inoculum. Data information: Data shown from a single experiment. Technical replicates $n = 4$ for (**A**), and $n = 3$ for (**B**).

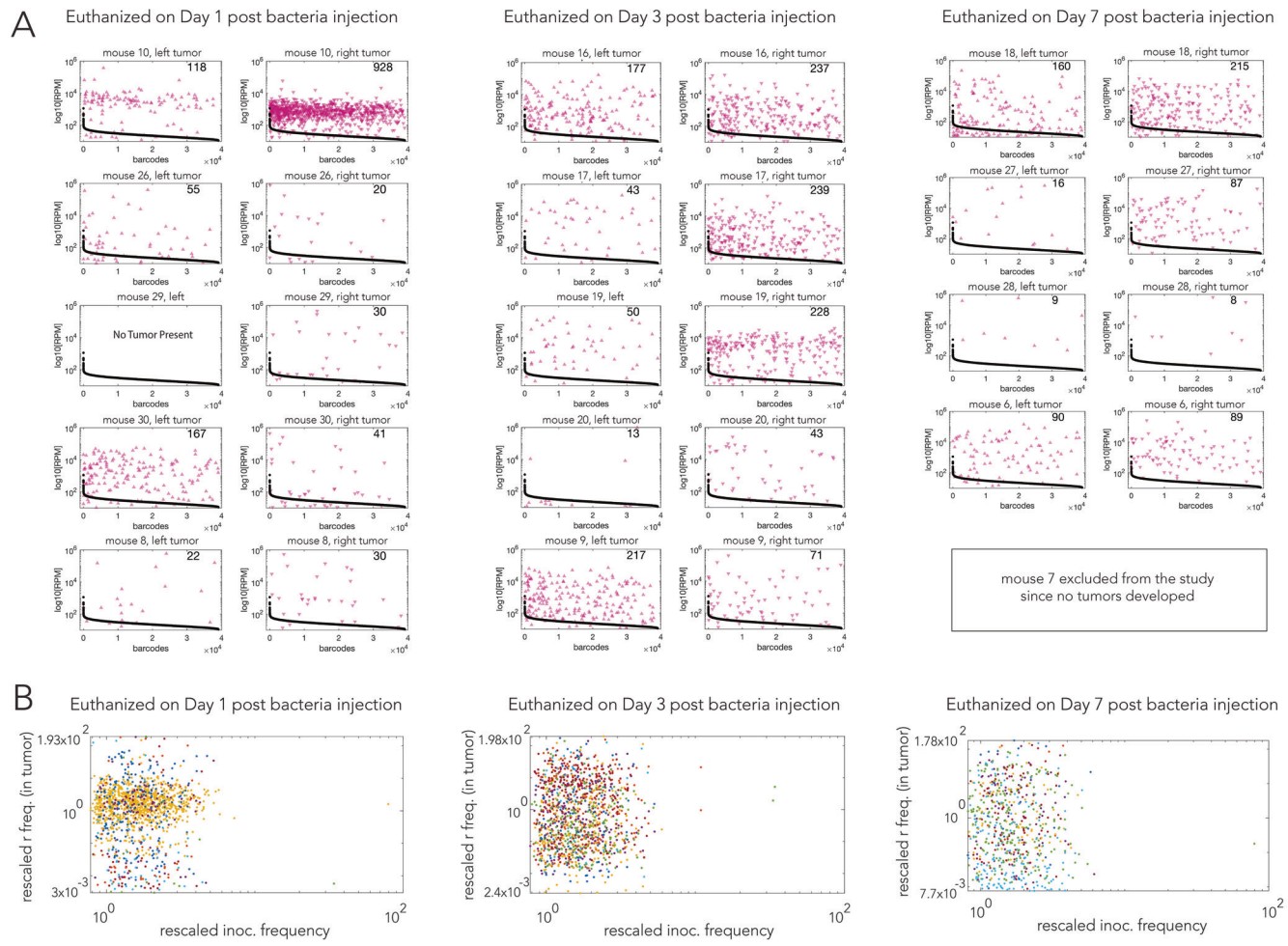

**Figure EV2.** (A) Frequency of detected unique barcodes from left and right tumors of individual mice from the i.v. injected condition grouped by day of euthanasia. Detected barcodes were ranked in decreasing frequency in the inoculum. The number in the upper right corner of each panel indicates the total number of unique barcodes in the tumor. (B) No correlation is found between the inoculum frequency and the frequency post i.v. injection for all days as shown in the last panel of each column (points in different colors mark different tumors). Spearman correlation coefficients were all around 0 [−0.02–0.02]. Data information: Data from a single experiment. Number of total mice = 14, technical replicates n = 2.

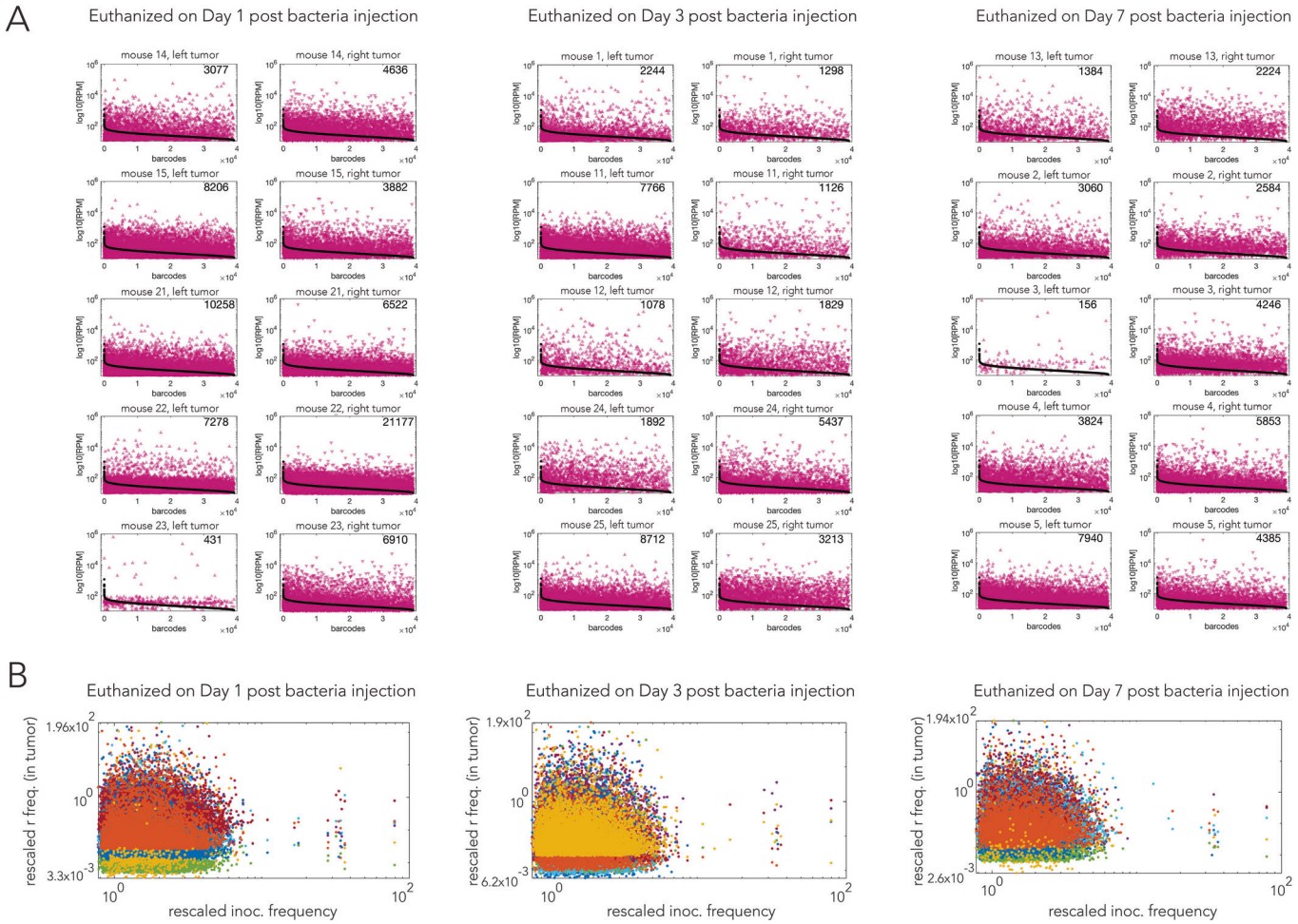

**Figure EV3.** **(A)** Frequency of detected unique barcodes from left and right tumors of individual mice from the i.t. injected condition, grouped by day of euthanasia. Detected barcodes were ranked by decreasing frequency in the inoculum. The number in the upper right corner of each panel indicates the total number of unique barcodes in the tumor. Number of total mice = 14, technical replicates $n = 2$. **(B)** No correlation is found between the inoculum frequency and the frequency post i.t. injection for all days, as shown in the last panel of each column (points in different colors mark different tumors). Spearman correlation coefficients were all around 0 [−0.02–0.02]. Data information: Data from a single experiment. Number of total mice = 15, technical replicates $n = 2$.

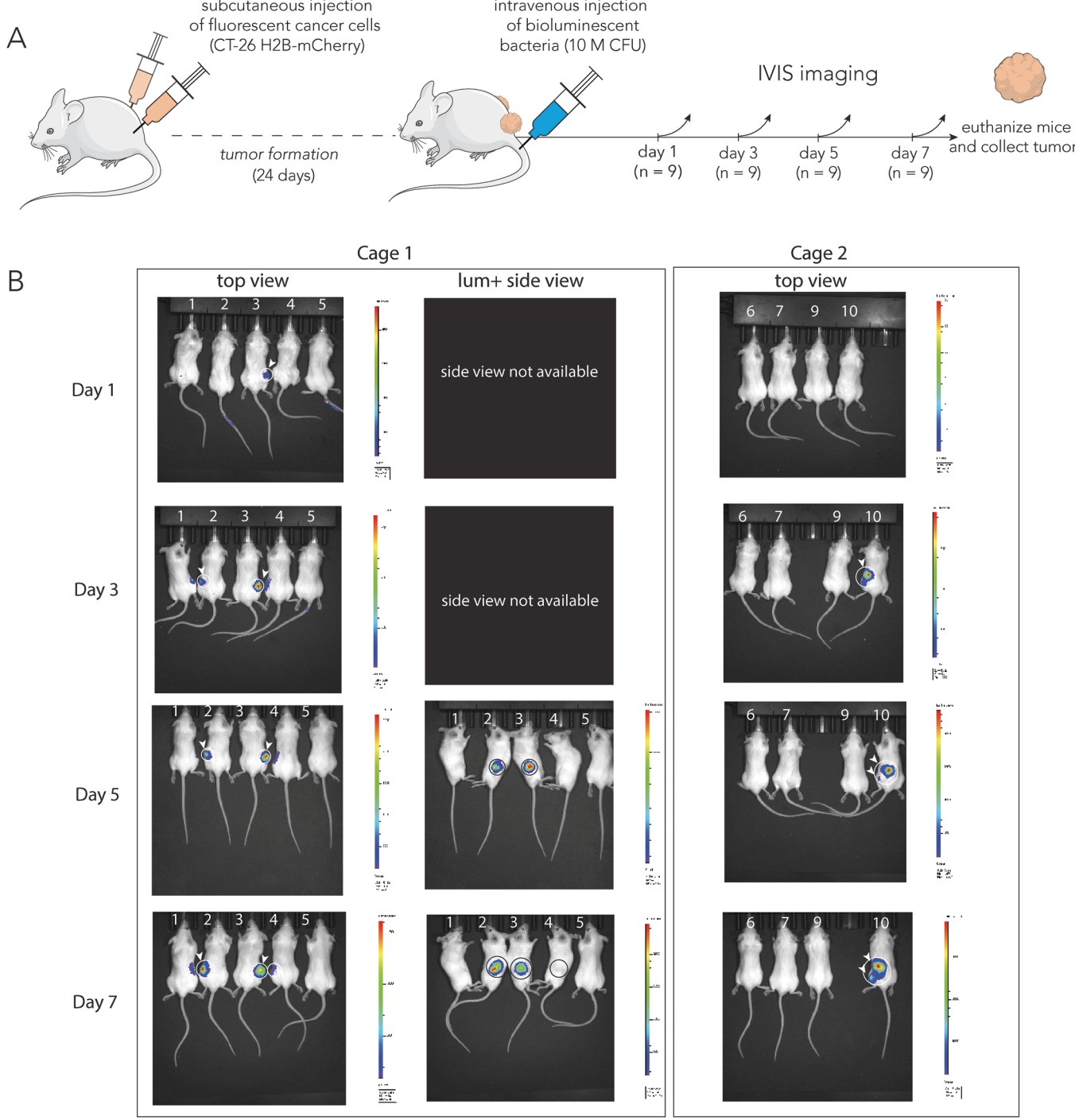

**Figure EV4. Bacteria localize exclusively to tumors.**

(A) Outline of experimental approach. Subcutaneous tumors were formed on the right and left flanks by injection of CT-26 cells and allowed to grow for 24 days. Tumors developed in four of the nine mice. All mice were injected intravenously with ten million CFUs of bioluminescent *E. coli* Nissle 1917. (B) Bioluminescent images of the mice were captured on days 1, 3, 5, and 7 post bacteria injection. Circles mark the location of developed tumors, and arrows mark a clear bioluminescent signal. A bioluminescent signal was observed exclusively in the tumor area (3 of 4 mice). In one mouse with a tumor, mice 4 of cage 1, bacterial colonization failed (1 of 4). We did not observe any bacteria in other organs in these mice (besides the initial wound from tail injection on day 1), and we did not observe bacteria in mice that did not have a developed tumor (5 of 5). Data information: Data from a single experiment. Number of total mice = 9.

