## [Peer Review File · Molecular Systems Biology]

Bacterial population dynamics during colonization of solid tumors

Serkan Sayin, Motasem ElGamel, Brittany Rosener, Michael Brehm, Andrew Mugler, and Amir Mitchell

Corresponding author(s): Amir Mitchell (amir.mitchell@umassmed.edu) , Andrew Mugler (andrew.mugler@pitt.edu)

Review Timeline:

Submission Date:	8th Jul 25
Editorial Decision:	27th Aug 25
Revision Received:	19th Oct 25
Editorial Decision:	18th Nov 25
Revision Received:	23rd Nov 25
Accepted:	24th Nov 25

Editor: Jingyi Hou

Transaction Report:

27th Aug 2025

Manuscript Number: MSB-2025-13224

Title: Bacterial population dynamics during colonization of solid tumors

Author: Serkan Sayin

Motasem ElGamel

Brittany Rosener

Michael Brehm

Andrew Mugler

Amir Mitchell

Dear Dr Mitchell,

Thank you for submitting your work to Molecular Systems Biology. First of all, I would like to apologise for the slow process. We have now heard back from the three reviewers who agreed to evaluate your manuscript. As you will see from the comments below that they find the manuscript to be of interest. They raise, however, several important points, which should be convincingly addressed in a revision of this work.

I think that the recommendations of the reviewers are rather clear so there is no need to repeat the points listed below. All issues raised by the reviewers need to be satisfactorily addressed. As you may already know, our editorial policy allows in principle a single round of major revision so it is essential to provide responses to the reviewers' comments that are as complete as possible. Please feel free to contact me in case you would like to discuss in further detail any of the issues raised by the reviewers.

On a more editorial level, we would ask you to address the following issues:

- Please provide a .docx formatted version of the manuscript text (including legends for main figures, EV figures and tables). Please make sure that the changes are highlighted to be clearly visible.
- Please provide individual production quality figure files as .eps, .tif, .jpg (one file per figure).
- Please provide a .docx formatted letter INCLUDING the reviewers' reports and your detailed point-by-point responses to their comments. As part of the EMBO Press transparent editorial process, the point-by-point response is part of the Review Process File (RPF), which will be published alongside your paper.
- Please note that all corresponding authors are required to supply an ORCID ID for their name upon submission of a revised manuscript.
- We replaced Supplementary Information with Expanded View (EV) Figures and Tables that are collapsible/expandable online (see examples in <http://msb.embopress.org/content/11/6/812>). A maximum of 5 EV Figures can be typeset. EV Figures should be cited as 'Figure EV1, Figure EV2' etc... in the text and their respective legends should be included in the main text after the legends of regular figures.

Additional Tables/Datasets should be labeled and referred to as Table EV1, Dataset EV1, etc. Legends have to be provided in a separate tab in case of .xls files. Alternatively, the legend can be supplied as a separate text file (README) and zipped together with the Table/Dataset file.

For the figures and tables that you do NOT wish to display as Expanded View figures, they should be bundled together with their legends in a single PDF file called *Appendix*, which should start with a short Table of Content. Each legend should be below the corresponding Figure/Table in the Appendix. Appendix figures and tables should be referred to in the main text as: "Appendix Figure S1, Appendix Figure S2, Appendix Table S1" etc. See detailed instructions regarding expanded view here: <https://www.embopress.org/page/journal/17444292/authorguide#expandedview>.

- Before submitting your revision, primary datasets (and computer code, where appropriate) produced in this study need to be deposited in an appropriate public database (see <http://msb.embopress.org/authorguide-dataavailability> <https://www.embopress.org/page/journal/17444292/authorguide#dataavailability>).

The accession numbers and database should be listed in a formal "Data Availability" section (placed after Materials & Method) that follows the model below (see also <https://www.embopress.org/page/journal/17444292/authorguide#dataavailability>). Please note that the Data Availability Section is restricted to new primary data that are part of this study.

Data availability

- At EMBO Press we ask authors to provide source data for the main manuscript figures. You will receive a separate email with instructions for providing source data with your revised manuscript, including how to upload and organize the files. Additional information on source data and instruction on how to label the files are available

- Our journal encourages inclusion of *data citations in the reference list* to directly cite datasets that were re-used and obtained from public databases. Data citations in the article text are distinct from normal bibliographical citations and should directly link to the database records from which the data can be accessed. In the main text, data citations are formatted as follows: "Data ref: Smith et al, 2001". In the Reference list, data citations must be labeled with "[DATASET]". A data reference must provide the database name, accession number/identifiers and a resolvable link to the landing page from which the data can be accessed at the end of the reference. Further instructions are available at .

- We updated our journal's competing interests policy in January 2022 and request authors to consider both actual and perceived competing interests. Please review the policy <https://www.embopress.org/competing-interests> and update your competing interests if necessary. Please use the heading "Disclosure statement and competing interests".

- All Materials and Methods need to be described in the main text using our 'Structured Methods' format. According to this format, the Methods section includes a Reagents and Tools Table (listing key reagents, experimental models, software and relevant equipment and including their sources and relevant identifiers) followed by a Methods and Protocols section describing the methods, ideally using a step-by-step protocol format. The aim is to facilitate adoption of the methodologies across labs.

Please download and fill our Reagents and Tools Table template (.docx), which you can find in our author guidelines: <https://www.embopress.org/page/journal/17444292/authorguide#structuredmethods>.

An example of a Method paper with Structured Methods can be found here: <https://www.embopress.org/doi/10.15252/msb.20178071>.

-Regarding data quantification:

Please ensure to specify the name of the statistical test used to generate error bars and P values, the number (n) of independent experiments (please specify technical or biological replicates) underlying each data point and the test used to calculate p-values in each figure legend. Discussion of statistical methodology can be reported in the materials and methods section, but figure legends should contain a basic description of n, P and the test applied.

Graphs must include a description of the bars and the error bars (s.d., s.e.m.).

- Please provide a "standfirst text" summarizing the study in one or two sentences (approximately 250 characters, including space), three to four "bullet points" highlighting the main findings and a "synopsis image" (550px width and 400-600 px height, PNG format) to highlight the paper on our homepage.

Here are a couple of examples:

<https://www.embopress.org/doi/10.15252/msb.20199356>

<https://www.embopress.org/doi/10.15252/msb.20209475>

<https://www.embopress.org/doi/10.15252/msb.209495>

When you resubmit your manuscript, please download our CHECKLIST (<https://www.embopress.org/pb-assets/embosite/EMBO%20Press%20Author%20Checklist-1642513524327.xlsx>) and include the completed form in your submission.

Please note that the Author Checklist will be published alongside the paper as part of the transparent process (<https://www.embopress.org/page/journal/17444292/authorguide#transparentprocess>).

If you feel you can satisfactorily deal with these points and those listed by the referees, you may wish to submit a revised version of your manuscript. Please attach a covering letter giving details of the way in which you have handled each of the points raised by the referees. A revised manuscript will be once again subject to review and you probably understand that we can give you no guarantee at this stage that the eventual outcome will be favorable.

I look forward to receiving the revised manuscript soon.

Kind regards,
Jingyi

Jingyi Hou, PhD
Senior Editor
Molecular Systems Biology

We realize that it is difficult to revise to a specific deadline. In the interest of protecting the conceptual advance provided by the work, we recommend a revision within 3 months (25th Nov 2025). Please discuss the revision progress ahead of this time with the editor if you require more time to complete the revisions. Use the link below to submit your revision:

*** PLEASE NOTE *** As part of the EMBO Press transparent editorial process initiative (see our Editorial at <https://dx.doi.org/10.1038/msb.2010.72>), Molecular Systems Biology publishes online a Review Process File with each accepted manuscripts. This file will be published in conjunction with your paper and will include the anonymous referee reports, your point-by-point response and all pertinent correspondence relating to the manuscript. If you do NOT want this File to be published, please inform the editorial office at contact@molsystbiol.org within 14 days upon receipt of the present letter.

Reviewer #1:

The authors use an isogenic barcoded population of E. coli (nissle strain) to track tumour colonisation dynamics in mice. This is a model for bacterial infection in tumour (similar colonisation experiment to ref 18 Geller et al, and engineered tumour therapeutics in refs) making it relevant. The question of mechanisms of colonisation and population dynamics is understudied therefore is relevant. The authors generate high quality data, but at this current version the analysis there are important controls needed which are critical for result interpretation.

First, while barcoded libraries can be isogenic in time zero, it has been observed that they can be highly evolving in the context of host colonisation, with changes in barcode frequency reflecting emergence of mutations under selection. E.g. for time zero isogenic barcoded population of E coli in mice gut showed that barcode selection reflected mutations in motility and metabolite utilization genes that helped colonisation (Vasquez et al., 2021) even in <7 days.

In the current manuscript, the modelling assumes all barcodes are genetically identical, thus equal fitness and nutritional preferences, and then interprets variations in barcodes dynamics as reflective of substrate availability in tumour (some barcode in high nutrient region, some not?). While this is plausible, it could be more likely that the emergence of mutations explains differences between barcodes. Therefore, before doing this modelling, there should be a barcoded dynamic assessment for evidence of strain selection and population sequencing (see comment below). Importantly, the current manuscript lacks any in situ localisation evidence of colonisation as e.g. microscopy with bacteria staining/reporter, to show that the colonisation indeed is patchy as assumed in the resource model. Therefore, with data re-analysis and additional evidence to support assumptions, this manuscript could be valuable addition. The supporting tables were not accessible in the reviewers' documents, so they haven't been assessed.

Major comments

1)The current analysis and interpretation of results assumes no mutations can emerge during colonisation, which is a questionable assumption. Towards addressing this assumption:
Barcode specific dynamics are currently not explored in Fig2,3, and the possibility of strain evolution not addressed. The classic graphic would $y = \text{barcode relative abundance}$ $x = \text{time}$, as in (Vasquez et al., 2021) . This is also helpful to observe if there is evidence of e.g. strong selection of barcodes which could be driven by emerging mutations, with strains taking over larger fragments of population over time. This is particularly important in cases as mentioned in line 183 "with leading clones taking up to half of the total bacterial load". In those cases, the gDNA used for barcode sequencing could be analysed for whole genome sequencing to detect possible SNPs or other genetic mutations. This is also the follow up experiment for mutations as in (Vasquez et al., 2021) , where barcode selection was indicative of emerging mutations in motility and metabolism genes .
Barcodes taking over could also explain the drop in total barcodes in day 7, drop in Shannon diversity, and differences between number of colonisation clones and total bacteria load. Thus, there are more systematic analysis that can be done to explore

experimental barcode dynamics. This also can be instead of illustrating clonal distribution in a single mice example as in Figure 2F; 3E; This analysis could be more comprehensive and take size dynamic per barcode over time; and observed trajectories.

2) There is a lack of evidence of the spatial localisation of *E. coli* nissle in tumour shown by imaging. While the set up is similar to Geller et al, there should still be evidence of preferent colonisation in tumour. This makes interpretation limiting considering that e.g. blood carry over of bacteria is not being analysed, which would be key to interpret Fig 2 analysis (how to tell apart tissue colonisation from cells circulating? This is not addressed). Considering the modelling is adapted to local niches model to fit distribution, any imaging evidence to support the local niche hypothesis would make this manuscript stronger.

3) Author's find the Zipf's law distribution unexpected, in line 423 "Mutations can give rise to subpopulations whose sizes can be distributed according to Zipf's law, as in the classic Luria-Delbruck process(...) These systems are distinct from ours in that new populations arise at various points in time or space due to mutations or immigration from a repertoire, whereas in our system no new barcodes emerge.". Emergence of genomic mutations at a distant locus would not be reflected as a new barcode, but rather a barcode changing its relative fitness. Therefore, the interpretation of Zipf's law as product of niche limitation needs more support, or at least to further investigate alternative genetic model that could explain Zipf's law.

Minor comments

- In general, the figures are missing the sample size and definitions of the statistics shown. Considering that it is not consistent in figures in a panel, this makes it hard to track. E.g. Figure S1 replicate space is not mentioned in figure legend. Also it is not specified what modelling is being compared against.

- Time zero values in Fig2B count axis should not be plotted as true data points if these are the expected inoculum value (which is already shown as dashed lines for comparison). In any case, if y axis is the count of bacteria in tumour, T0 of infection would be expected to be zero?

- Figure FIG2b mentions 39 barcodes. It should be 39k (39,000).

- Figure 2D and 3D could just plot the inner graphic, there is no need to make the axis equal in scale if there are two different units/variables.

- Figure 2C and 3C plot the identity line instead of the model fit.

- Titles of result section could better reflect the results rather than the method.

- Wording when referring to figures can be unclear e.g. line 137 "As the pink graph shows,..." (many graphs are pink)

Vasquez, K. S., Willis, L., Cira, N. J., Ng, K. M., Pedro, M. F., Aranda-Díaz, A., Rajendram, M., Yu, F. B., Higginbottom, S. K., Neff, N., Sherlock, G., Xavier, K. B., Quake, S. R., Sonnenburg, J. L., Good, B. H., & Huang, K. C. (2021). Quantifying rapid bacterial evolution and transmission within the mouse intestine. *Cell Host & Microbe*, 29(9), 1454-1468.e4. <https://doi.org/10.1016/J.CHOM.2021.08.003>

Reviewer #2:

In Sayin & ElGamel et al., the authors created a library of genetically identical *E. coli* strains (*E. coli* Nissle), each carrying a unique DNA barcode, and used it to challenge mice with subcutaneous tumors via intravenous or direct intratumor injection. By sequencing these barcodes from resected tumors at different time points, they tracked the fate and expansion of individual clones. Their results showed that colonization is shaped by a big bottleneck with only a small number of bacterial clones successfully colonising each tumor. This study provides a framework to understand bacterial colonization dynamics in tumors, highlighting the role of bottlenecks and stochastic growth in shaping tumor populations. Its focus on a single genetically uniform bacterial strain limits findings into functional diversity and microbial interactions that could occur in natural tumor microbiomes. Nonetheless, the integration of experimental lineage tracking with mathematical modeling offers valuable tools for future research in tumor microbiology.

Comments:

-Line 160: "and the number of bacteria seeding the infection is independent from the total bacterial load observed after colonization". How was this proven? This would require to inject different amount of bacteria and measure the bacterial load at the tumor side. Same in line 213.

-Line 197: "Thus, similarly to intravenous injection, tumor pairs in the intratumor injection model seem to have operated as separate and independent niches (supplementary table 1)." Their results show that bacteria colonising the tumor pairs are different, and that the each tumor acts as separate and independent niche. But what about the similarity or difference between the different mice? Is there a bacterial signature for each tumor side or is it just all random?

-While I understand the use of genetically identical clones in the library, a knockout library would give a lot more functional information, like showing which genes are necessary for bacteria to colonise and survive in the tumor. This should be at least discuss in the discussion.

- The explanation of what has been done at the end of the introduction seems too much (line 65-91). I would advise to shorten it and remove the mention of Figures in the introduction.
- Line 94: I would not name the section "Cloning the E. coli barcoded strain collection" as it reads like a section of the methods. Instead, use a title that describes the main findings of this section. Same actually for all the results sections.
- Tumors are not "infected" if an infection is not develop. The correct term here is either colonised or challenged.
- Fig 3E shows results from a representative single mouse. How representative is it? Can authors put the data for the other mice in the supplementary?

Reviewer #3:

In this paper, Sayin et al. use barcoded E. coli Nissle colonization in a mouse tumor model via both intravenous and intratumoral infection to model tumor microbiome colonization dynamics. They apply a consumer-resource modeling framework constrained by infection bottlenecks, local niche load, and stochastic noise to interpret colonization outcomes.

Overall, this paper is informative for the cancer microbiome field, particularly in providing a mechanistic perspective on microbial colonization within the tumor microenvironment. Despite persisting ambiguities regarding the very existence and functional role of the "tumor microbiome," the modeling approach is valuable and the work is overall worthy of publication. However, I have some major concerns about the representation of data and interpretation of results, which I outline below.

Major Comments

1. The priority effects observed in vitro raise the question of whether the same dynamics apply in vivo. A subset of barcodes exceeds the 99% diversity threshold. Do the successful colonizers in tumors largely come from this set? Is there a correlation between initial abundance and colonization outcomes across intravenous and intratumoral infections? A correlation plot would make the claim about non-correlated barcode frequencies across mice more convincing.
2. Please include r- and p-values directly on the graphs so readers can interpret statistical strength without needing to refer back to the text.
3. The assumption in Lines 183-186 that colonization arises from a single cell is overly simplistic. It would be more accurate to estimate the number of founding clones probabilistically by comparing the relative proportions in the inoculum and endpoint. This approach would yield a more realistic generation time and aligns with later results on probability distributions.
4. Entropy trends after day 1 should be clarified. If entropy continues to fluctuate beyond this point, does that mean the system never fully stabilizes? This seems important for interpreting Figure 2G. Authors should clarify that in the text.
5. The ranked frequency distributions appear visually consistent with Zipf's law, but this claim would be stronger if correlation coefficients were calculated and reported, confirming the statistical fit.
6. The assumption that growth is limited by each clone's own abundance may not hold in this experimental context. Once niche selection has occurred, surviving clones should compete for shared resources. Has the model been tested using a formulation where growth depends on total abundance (i.e., K scaled by the sum of all clones)? Would Zipf's law still emerge in that case?

Minor Comments

1. Figures should be consistently referenced by number rather than descriptions such as "pink graphs" (e.g., Line 137, and elsewhere).
2. In Fig. 1B, explicitly label the alternative infection route for clarity.
3. Line 160: refer to the number of bacterial clones rather than "number of bacteria."
4. Several sentences in the Results section are unnecessarily repeated (e.g., Lines 195-197 and again beginning at Line 206). Please consolidate these to improve readability.

Reviewer #1:

The authors use an isogenic barcoded population of *E. coli* (nissle strain) to track tumour colonisation dynamics in mice. This is a model for bacterial infection in tumour (similar colonisation experiment to ref 18 Geller et al, and engineered tumour therapeutics in refs) making it relevant. The question of mechanisms of colonisation and population dynamics is understudied therefore is relevant. The authors generate high quality data, but at this current version the analysis there are important controls needed which are critical for result interpretation.

First, while barcoded libraries can be isogenic in time zero, it has been observed that they can be highly evolving in the context of host colonisation, with changes in barcode frequency reflecting emergence of mutations under selection. E.g. for time zero isogenic barcoded population of *E. coli* in mice gut showed that barcode selection reflected mutations in motility and metabolite utilization genes that helped colonisation (Vasquez et al., 2021) even in <7 days. In the current manuscript, the modelling assumes all barcodes are genetically identical, thus equal fitness and nutritional preferences, and then interprets variations in barcodes dynamics as reflective of substrate availability in tumour (some barcode in high nutrient region, some not?). While this is plausible, it could be more likely that the emergence of mutations explains differences between barcodes. Therefore, before doing this modelling, there should be a barcoded dynamic assessment for evidence of strain selection and population sequencing (see comment below). Importantly, the current manuscript lacks any in situ localisation evidence of colonisation as e.g. microscopy with bacteria staining/reporter, to show that the colonisation indeed is patchy as assumed in the resource model. Therefore, with data re-analysis and additional evidence to support assumptions, this manuscript could be valuable addition. The supporting tables were not accessible in the reviewers' documents, so they haven't been assessed.

Major comments

1)The current analysis and interpretation of results assumes no mutations can emerge during colonisation, which is a questionable assumption. Towards addressing this assumption: Barcode specific dynamics are currently not explored in Fig2,3, and the possibility of strain evolution not addressed. The classic graphic would $y = \text{barcode relative abundance}$ $x = \text{time}$, as in (Vasquez et al., 2021) . This is also helpful to observe if there is evidence of e.g. strong selection of barcodes which could be driven by emerging mutations, with strains taking over larger fragments of population over time. This is particularly important in cases as mentioned in line 183 "with leading clones taking up to half of the total bacterial load". In those cases, the gDNA used for barcode sequencing could be analysed for whole genome sequencing to detect possible SNPs or other genetic mutations. This is also the follow up experiment for mutations as in (Vasquez et al., 2021) , where barcode selection was indicative of emerging mutations in motility and metabolism genes .

Barcodes taking over could also explain the drop in total barcodes in day 7, drop in Shannon diversity, and differences between number of colonisation clones and total bacteria load. Thus, there are more systematic analysis that can be done to explore experimental barcode dynamics.

This also can be instead of illustrating clonal distribution in a single mice example as in Figure 2F; 3E; This analysis could be more comprehensive and take size dynamic per barcode over time; and observed trajectories.

We are familiar with the Vasquez paper that reviewer referred to. However, there is one important difference between the two experimental systems, while the Vasquez model allow for longitudinal sampling of collection from the same animal, sampling in our animal model is performed by euthanizing the mice and extracting the tumors making it impossible to monitor how barcodes change over time in a single tumor. However, we did expand the results. We provided barcode frequency distributions for all mice in new extended view Figures 2 and 3. In addition, we performed One way ANOVA on Shannon Entropy results and did not find statistically significant difference between Days 1, 3 and 7. This suggests that after the initial drop, barcode diversity remains stable throughout the experiment.

We understand the reviewer's concern about spontaneous advantageous mutations can potentially underly tumor take-over. We directly tested this possibility by whole genome sequencing of the 10 clones isolated from different tumors, in which eight were the most dominant clone in the tumor they were isolated from (the other two barcodes were highly frequent, but not the most dominant). We did not identify any mutations in these clones (beyond recovery of the barcode sequences themselves). We added report about these observations in the results and methods sections and provided analysis results in Table EV3.

2) There is a lack of evidence of the spatial localisation of *E. coli* nissle in tumour shown by imaging. While the set up is similar to Geller et al, there should still be evidence of preferent colonisation in tumour. This makes interpretation limiting considering that e.g. blood carry over of bacteria is not being analysed, which would be key to interpret Fig 2 analysis (how to tell apart tissue colonisation from cells circulating? This is not addressed). Considering the modelling is adapted to local niches model to fit distribution, any imaging evidence to support the local niche hypothesis would make this manuscript stronger.

Our animal model is identical to that used by multiple other researchers in the field and we had no reason to doubt bacterial localization. High number of circulating bacteria are expected to lead to bacteremia and adverse host effects associated with sepsis (none were evident in our mice). To directly address the reviewer comment we designed an experiment to monitor bacteria localization in tumors using IVIS imaging (similarly to that used by Geller at al.) and included that results in extended view Figure 4. Briefly, This experiment uses the same mouse model and bioluminescent *E. coli* Nissle1917 strain injected intravenously. IVIS spectrum imaging across multiple days post bacteria injection showed tumor localization only in mice possessing measurable tumors (3 of 4 mice). In one mouse with a tumor bacterial colonization failed (1 of 4). We did not observe any bacteria in other organs in these mice (beside the initial wound from tail injection in day 1) and we did not observe bacteria in any mouse that did not have a measurable tumor (5 of 5)

3) Author's find the Zipf's law distribution unexpected, in line 423 "Mutations can give rise to subpopulations whose sizes can be distributed according to Zipf's law, as in the classic Luria-Delbruck process(...) These systems are distinct from ours in that new populations arise at various points in time or space due to mutations or immigration from a repertoire, whereas in our system no new barcodes emerge.". Emergence of genomic mutations at a distant locus would not be reflected as a new barcode, but rather a barcode changing its relative fitness. Therefore, the interpretation of Zipf's law as product of niche limitation needs more support, or at least to further investigate alternative genetic model that could explain Zipf's law.

We explored a few alternative models using theory and they do not give rise to Zipf's law. Whole genome sequencing ruled out spontaneous advantageous mutations as key mechanism underlying barcode dominance. Taking these two together, we think this is sufficient to support our claim.

Minor comments

- In general, the figures are missing the sample size and definitions of the statistics shown. Considering that it is not consistent in figures in a panel, this makes it hard to track. E.g. Figure S1 replicate space is not mentioned in figure legend. Also it is not specified what modelling is being compared against.

We added sample size (biological replicates, technical replicates), and statistic tests to the figure legends. We made an error and included only in-vitro data in Fig. S1 (now EV1). We corrected the error and Fig. EV1 now shows a comparison between the in-vitro data and model predictions in the in-vitro limit obtained by simulating Eqs. 14 and 15.

- Time zero values in Fig2B count axis should not be plotted as true data points if these are the expected inoculum value (which is already shown as dashed lines for comparison). In any case, if y axis is the count of bacteria in tumour, T0 of infection would be expected to be zero?

This was corrected. However, since y axis is logarithmic and $\log[0]$ is $-\infty$, only Days 1,3 and 7 are shown in the figure now.

- Figure FIG2b mentions 39 barcodes. It should be 39k (39,000).

Corrected

- Figure 2D and 3D could just plot the inner graphic, there is no need to make the axis equal in scale if there are two different units/variables.

Corrected

- Figure 2C and 3C plot the identity line instead of the model fit.

2C and 3C already plots the identity line $y=x$

- Titles of result section could better reflect the results rather than the method.

Titles were edited to describe the results.

- Wording when referring to figures can be unclear e.g. line 137 "As the pink graph shows,..." (many graphs are pink)

Corrected.

Vasquez, K. S., Willis, L., Cira, N. J., Ng, K. M., Pedro, M. F., Aranda-Díaz, A., Rajendram, M., Yu, F. B., Higginbottom, S. K., Neff, N., Sherlock, G., Xavier, K. B., Quake, S. R., Sonnenburg, J. L., Good, B. H., & Huang, K. C. (2021). Quantifying rapid bacterial evolution and transmission within the mouse intestine. *Cell Host & Microbe*, 29(9), 1454-1468.e4. <https://doi.org/10.1016/J.CHOM.2021.08.003>

Reviewer #2:

In Sayin & ElGamel et al., the authors created a library of genetically identical E. coli strains (E. coli Nissle), each carrying a unique DNA barcode, and used it to challenge mice with subcutaneous tumors via intravenous or direct intratumor injection. By sequencing these barcodes from resected tumors at different time points, they tracked the fate and expansion of individual clones. Their results showed that colonization is shaped by a big bottleneck with only a small number of bacterial clones successfully colonising each tumor. This study provides a framework to understand bacterial colonization dynamics in tumors, highlighting the role of bottlenecks and stochastic growth in shaping tumor populations. Its focus on a single genetically uniform bacterial strain limits findings into functional diversity and microbial interactions that could occur in natural tumor microbiomes. Nonetheless, the integration of experimental lineage tracking with mathematical modeling offers valuable tools for future research in tumor microbiology.

Comments:

-Line 160: "and the number of bacteria seeding the infection is independent from the total bacterial load observed after colonization". How was this proven? This would require to inject different amount of bacteria and measure the bacterial load at the tumor side. Same in line 213.

This statements about independence are conclusions stemming from the observation that there is no significant correlation between the number of barcodes in the tumor (assumed to represent the number of bacteria colonizers) and the total bacterial load in the tumor. We added a clarifying sentence about this conclusion in the text ("Assuming that each detected barcode represents a successful colonization event, we can infer that the tumor bacterial load does not depend on the initial number of colonizers.")

-Line 197: "Thus, similarly to intravenous injection, tumor pairs in the intratumor injection model seem to have operated as separate and independent niches (supplementary table 1)." Their results show that bacteria colonising the tumor pairs are different, and that the each tumor acts as separate and independent niche. But what about the similarity or difference between the different mice? Is there a bacterial signature for each tumor side or is it just all random?

Yes, niche independence holds across mice and across tumor sides as well. We added sentence about this lack of significance in the text and added a supplementary table EV4 with all the correlation results.

-While I understand the use of genetically identical clones in the library, a knockout library would give a lot more functional information, like showing which genes are necessary for bacteria to colonise and survive in the tumor. This should be at least discuss in the discussion.

We understand the importance of performing similar experiment using a knockout library to find identify gene required for colonization. In fact, our original study started off by performing such experiments using a barcoded knockout library. However, due to the presence of colonization bottlenecks and uneven growth, the noise introduced by population dynamics was far greater than the biological signal (the fitness effects of single gene knockouts). This realization lead us to the work presented in the manuscript. We thank reviewer for this raising this point and we added comments about it in the discussion section.

-The explanation of what is has been done at the end of the introduction seems too much (line 65-91). I would advise to shorten it and remove the mention of Figures in the introduction.

As the reviewer suggested, we shortened the relevant section in the end of the introduction. We decided however to mention Figures 1A and 1B since these conceptual diagrams are important for framing the key open question our work addresses.

-Line 94: I would not name the section "Cloning the E. coli barcoded strain collection" as it reads like a section of the methods. Instead, use a title that describes the main findings of this section. Same actually for all the results sections.

We revised this header and a few other so they reflect the main finding.

-Tumors are not "infected" if an infection is not develop. The correct term here is either colonised or challenged.

Infection term was corrected throughout the paper to colonization.

-Fig 3E shows results from a representative single mouse. How representative is it? Can authors put the data for the other mice in the supplementary?

Data from other mice (both i.v. and i.t.) provided as Expanded View Figure 2 and Expanded View Figure 3 respectively.

Reviewer #3:

In this paper, Sayin et al. use barcoded *E. coli* Nissle colonization in a mouse tumor model via both intravenous and intratumoral infection to model tumor microbiome colonization dynamics. They apply a consumer-resource modeling framework constrained by infection bottlenecks, local niche load, and stochastic noise to interpret colonization outcomes.

Overall, this paper is informative for the cancer microbiome field, particularly in providing a mechanistic perspective on microbial colonization within the tumor microenvironment. Despite persisting ambiguities regarding the very existence and functional role of the "tumor microbiome," the modeling approach is valuable and the work is overall worthy of publication. However, I have some major concerns about the representation of data and interpretation of results, which I outline below.

Major Comments

1. The priority effects observed in vitro raise the question of whether the same dynamics apply in vivo. A subset of barcodes exceeds the 99% diversity threshold. Do the successful colonizers in tumors largely come from this set? Is there a correlation between initial abundance and colonization outcomes across intravenous and intratumoral infections? A correlation plot would make the claim about non-correlated barcode frequencies across mice more convincing.

The reviewer raises two questions that are important to clarify. The first is whether barcode frequency in the inoculum is correlated with colonization outcomes (number of tumors that were successfully colonized by that specific barcode). And the second is whether barcode frequency in the inoculum correlated with the final frequency of barcodes (at collection from a tumor).

For the first question, we find that the data shows strong correlation between observation in the tumor, i.e., inoculum frequency is correlated with colonization success (shown in figures 5A and 5B). This observation is compatible with the model of each bacterial cell having the same probability to successfully colonize the tumor and that these events are independent from one another. We have clarified this point in the paper (lines 379-386).

To address the second question, we plotted the rescaled frequencies (frequencies/standard deviation) of the inoculum (x-axis) and post injection frequencies (y-axis) for the IT and IV cases. We found no correlation between barcode frequency in the inoculum and barcode frequency post injection. We added panels showing these frequency relationships to extended view figures 2 and 3 and we added a comment on this result in the paper (lines 386-387).

2. Please include r - and p -values directly on the graphs so readers can interpret statistical strength without needing to refer back to the text.

r and p values were added onto the panels Figure 2C, 2D, 2E and Figure 3C and 3D.

3. The assumption in Lines 183-186 that colonization arises from a single cell is overly simplistic. It would be more accurate to estimate the number of founding clones probabilistically

by comparing the relative proportions in the inoculum and endpoint. This approach would yield a more realistic generation time and aligns with later results on probability distributions.

We thank the reviewer for pointing this out. In the methods section, we show mathematically using our bottleneck model (Eq. (12)) and success probability of infection q found from data that the majority of cells in a barcode die with probability 77%. The probability of one cell in a barcode successfully seeding the tumor is 20%, while for two cells it is 2.6% (also included in the paper). These estimates are based on IT data. For IV data, probabilities of reaching the tumor will be even smaller. Thus, it is safe to assume that colonization in the tumor arises from only one representative cell of each barcode in most cases. In the rare case that a barcode is founded by two cells with that barcode, it will have a negligible effect on the calculated generation time (the number of generations can be corrected by subtracting one generation).

4. Entropy trends after day 1 should be clarified. If entropy continues to fluctuate beyond this point, does that mean the system never fully stabilizes? This seems important for interpreting Figure 2G. Authors should clarify that in the text.

From our mathematical understanding of the system we predict that the entropy saturates. A statistical test (one way ANOVA) does not find statistically significant differences among the means across days.

5. The ranked frequency distributions appear visually consistent with Zipf's law, but this claim would be stronger if correlation coefficients were calculated and reported, confirming the statistical fit.

We thank the reviewer for their suggestion. We fitted the rank-frequency IT data to a nonlinear function of the form $f = \beta r^{-\alpha}$ for all experiments. We obtained a best fit α value for each experiment. We then performed a Wilcoxon signed rank test on the best fit α data with the null hypothesis that α is sampled from a distribution whose median is 1. The test fails to reject the null hypothesis (p -value 0.2275). We reported the median and standard deviation of α in the paper (lines 240-243).

6. The assumption that growth is limited by each clone's own abundance may not hold in this experimental context. Once niche selection has occurred, surviving clones should compete for shared resources. Has the model been tested using a formulation where growth depends on total abundance (i.e., K scaled by the sum of all clones)? Would Zipf's law still emerge in that case?

We thank the reviewer for raising this important point. It is true that surviving clones should compete for shared resources which we allow in our model. In Eq. (4), which specify the resource dynamics, resource consumption is summed over all the surviving barcodes. The effect of niche limitation is a local effect added in addition to the normal resource competition.

We have tested our model with global growth limitation (i.e. replacing the term $s/(s + kN_i)$ with $s/(s + k\sum_i N_i)$), we found that this leads to only one barcode surviving similar to the Monod

model (Eqs. (1) and (2)). The reason is that by removing the local niche limitation and making it global we allowed large barcodes to dominate the population and have an advantage at resource consumption which leads to survival of only one barcode (in the absence of any additional resources that can be shared).

We have clarified this point in the paper (lines 342-346)

Minor Comments

1. Figures should be consistently referenced by number rather than descriptions such as "pink graphs" (e.g., Line 137, and elsewhere).

Statements referring to the pink lines were deleted. Figure number is referenced before commenting on the result.

2. In Fig. 1B, explicitly label the alternative infection route for clarity.

Infection routes were added to Figure 1B.

3. Line 160: refer to the number of bacterial clones rather than "number of bacteria."

In the intravenous model we assumed that the number of barcodes detected is almost identical to the number of bacteria seeding the tumor (see answer to Major point 3). We added a clarifying statement to the sentence.

4. Several sentences in the Results section are unnecessarily repeated (e.g., Lines 195-197 and again beginning at Line 206). Please consolidate these to improve readability.

These two sentences refer to different results. We revised the sentences to clarify this better. Lines 202-204: Statement refers to the lack of correlation in the frequencies of barcodes in same-mouse tumor pairs in any of the i.t. mice (a statistical test if performed on each mouse to test for significant correlation in frequencies)

213-216: Statement refer to the lack of correlation in the number of detected barcodes in the left flank of all mice compared to the right flank of all mice (a single statistical test is performed for the all 15 mice).

18th Nov 2025

Manuscript Number: MSB-2025-13224R
Title: Bacterial population dynamics during colonization of solid tumors
Author: Serkan Sayin
Motasem ElGamel
Brittany Rosener
Michael Brehm
Andrew Mugler
Amir Mitchell

Dear Dr Mitchell,

Thank you for the submission of your revised manuscript to Molecular Systems Biology. We have now received the enclosed report from the two reviewers who agreed to re-assess it. As you will see, the reviewers are now both supportive, and I am pleased to inform you that we will be able to accept your manuscript pending the following amendments:

1. Please address the remaining minor comment from Reviewer #1 regarding the request to add more discussion.

On a more editorial level, please do the following:

1. Add the corresponding authors' email addresses on the title page.
2. Remove the "Author Contribution" section from the manuscript file.
3. Provide up to five keywords in the manuscript file.
4. Add a "DISCLOSURE AND COMPETING INTERESTS STATEMENT." Employment in a biotech company should be disclosed in this section.
5. The provided synopsis image is too large. Please resubmit it in the required dimensions (550 px width and 400-600 px height, PNG format).
6. Source file names, titles, legends and manuscript callouts all need to be updated to Dataset EV1-EV4 instead of Tables EV1-EV4. Legends must be removed from the manuscript and uploaded as a separate tab/sheet within each Excel file.
7. "Materials and methods" should be renamed to "Methods".
8. EV table legends should be removed from manuscript file.
9. Ensure that all provided source data files are correctly labeled in the source data checklist.
 - Please add the missing checkmarks for Figures 1-3 in the source data checklist.
 - Please provide the missing source data for Figures 4G and 5F.
10. Sections need to be named and the order should be corrected: Title page - Abstract - Keywords - Introduction - Results - Discussion - Methods - Data Availability - Acknowledgements - Disclosure and Competing Interests Statement - References - Figure Legends - Table(s) - Expanded View Figure Legends.

Click on the link below to submit your revised paper.

Sincerely,
Jingyi

*** PLEASE NOTE *** As part of the EMBO Press transparent editorial process initiative (see our Editorial at <https://dx.doi.org/10.1038/msb.2010.72> , Molecular Systems Biology will publish online a Review Process File to accompany accepted manuscripts. When preparing your letter of response, please be aware that in the event of acceptance, your cover letter/point-by-point document will be included as part of this File, which will be available to the scientific community. More information about this initiative is available in our Instructions to Authors. If you have any questions about this initiative, please contact the editorial office (msb@embo.org).

Reviewer #1:

The authors have addressed the possibility of genetic selection by whole genome sequencing of colonies derived from *E. coli* Nissle dominant barcode strains. This strengthens the conclusion that barcode enrichment patterns are caused by factors other than genetic selections. Therefore, the characterisation of the bottleneck effect is very valuable, especially for informing future designs of genetic screens of tumour colonisation (as the authors mention on discussion).

The authors also provide additional support for specific tumour colonisation using luminescence. However, there is no direct imaging of the tumour tissue samples to support the proposed spatial separated bacterial colonisation patterns. It will be clarify this in the Discussion. They cite relevant literature where this type of localisation has been observed in bacteria colonised tumour tissue. It could help the readers outside the field to cite more examples of histology samples supporting this type of spatially isolated colonisation.

Overall, the manuscripts present broad dataset and analysis which will be helpful for the emerging field of bacteria-tumour colonisation. In this context, the discussion section would benefit from addressing the limitations of the study and potential future directions. Alternative factors that could explain local niches need to be explicitly discussed (e.g. priority effect, allee effect, differential immune response etc.).

Reviewer #3:

The authors have thoughtfully addressed all of the questions and concerns I raised in the previous round. I appreciate the effort that went into clarifying key points of all the reviewers' questions and strengthening both the analyses and the presentation. The revised manuscript reads more clearly, and the responses demonstrate careful consideration of the feedback.

I'm happy with the revisions and have no further concerns. I wish the authors all the best and encourage them to continue pursuing this line of research. It's exciting work and a meaningful contribution to the field.

All editorial and formatting issues were resolved by the authors.

24th Nov 2025

Manuscript number: MSB-2025-13224RR

Title: Bacterial population dynamics during colonization of solid tumors

Dear Dr Mitchell,

Thank you again for sending us your revised manuscript. We are now satisfied with the modifications made and I am pleased to inform you that your paper has been accepted for publication.

Sincerely,
Jingyi

Jingyi Hou, PhD
Senior Editor
Molecular Systems Biology
